# Towards ultra-sensitive and rapid near-source wastewater-based epidemiology

Da Huang [1] ✉, Alyssa Thomas DeCruz [1,2], Dounia Cherkaoui[1,2], Benjamin Miller[1] ✉, Diluka Peiris[1], Samuel Hopgood[1,2], Jessica Kevill[3], Kata Farkas[3], Rachel Williams[3], Davey L. Jones [3] & Rachel A. McKendry [1,2] ✉

Wastewater-based epidemiology is emerging as a powerful early-warning public health surveillance tool. However, gold-standard PCR necessitates transporting samples to laboratories, with significant reporting delays (24-72 h), prompting growing interest in rapid, near-source tests for resource-limited settings. Research has focused on gold nanoparticle dipsticks, but these typically lack sensitivity in wastewater. Herein, we explore two complementary nanomaterial based approaches, using SARS-CoV-2 as an exemplar: 1) visually-read carbon black dipsticks; 2) spin-enhanced fluorescent nanodiamond dipsticks, exploiting selective separation from background autofluorescence. The assay provides a 2-hour turnaround from sample preparation to result with minimal equipment and achieves a limit of detection down to 7 copies per assay. A pilot study with samples from the Welsh National WBE programme finds 80% sensitivity and 100% specificity for carbon black, and 100% sensitivity, specificity for nanodiamonds. A proof-of-concept lab-in-a-suitcase nanodiamond assay tests raw, unprocessed wastewater samples. These findings lay the foundations for near-source WBE early-warning quantum sensors in the environment.

Pathogenic viruses pose a major global public health threat, causing disease outbreaks that have significant human and economic consequences[1–3]. The global COVID-19 pandemic, caused by a previously unknown virus (SARS-CoV-2), has infected an estimated 775 million people and resulted in over 7.0 million reported deaths to date[4]. Moreover, in recent years, re-emerging poliovirus or norovirus have been detected in sewers, triggering major new vaccination and surveillance programmes[5–8]. The detection of microbial pathogens in wastewater has a long history dating back to 1870, when John Snow traced clusters of deaths from cholera to a contaminated pump on Broad Street London[9]. Today, more than 70 countries have adopted Wastewater-Based Epidemiology (WBE) an early warning tool for COVID-19, mpox, and many other pathogens[10–13].

For SARS-CoV-2 WBE, viral RNA fragments shed into faeces are detectable in wastewater as an early warning tool for outbreaks, to track temporal and geospatial changes, to clusters, and to guide prevention strategies, such as vaccination programmes[14]. A key advantage of WBE is that it can detect both symptomatic and asymptomatic cases within days of infection, even before clinic testing would take place, mitigating biases associated with surveillance from clinic attendance[15]. Moreover, WBE is capable of detecting outbreaks in communities that are often hard to reach due to social stigma, migrant or travelling communities, healthcare inequalities in access to infrastructural or resource limitations, and low healthcare utilisation regions[16,17]. Current techniques can be extremely sensitive; high-throughput polymerase-chain reaction (PCR) and sequencing has been used to report the detection of variants of concern in wastewater up to 14 days earlier than clinic-based surveillance[18,19]. Multiple incidences of viral spread not captured by clinical genomic surveillance were also reported[20–23] (Fig. 1a). The ability to detect and

[1]London Centre for Nanotechnology, University College London, London WC1H 0AH, UK. [2]Division of Medicine, University College London, London WC1E 6BT, UK. [3]School of Environmental and Natural Sciences, Bangor University, Bangor, Gwynedd LL57 2UW, UK. ✉e-mail: d.huang@ucl.ac.uk; ben.miller@ucl.ac.uk; r.a.mckendry@ucl.ac.uk

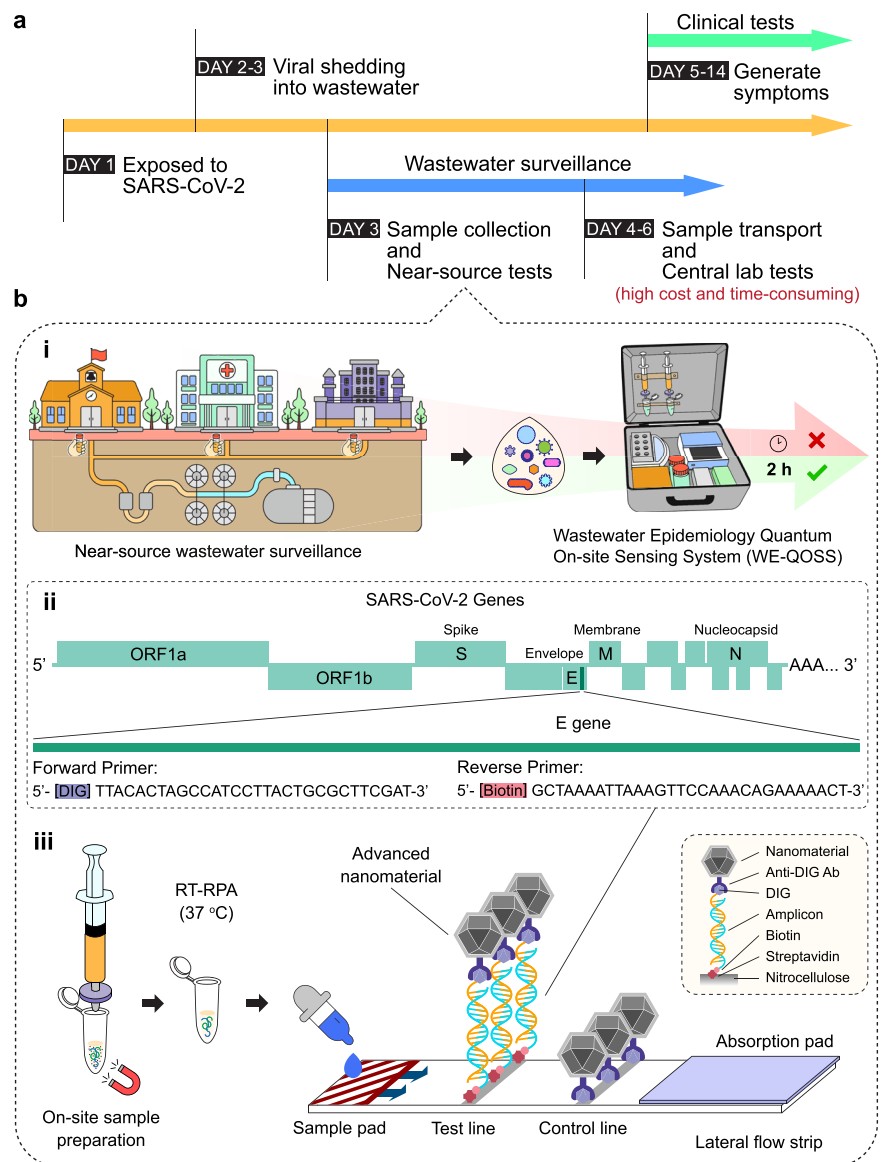

**Fig. 1 | Concept of a rapid near-source wastewater surveillance tool based on rapid dipstick assay for SARS-CoV-2 detection. a** A timeline illustrating the advantages of utilising wastewater pathogenic surveillance for early warning of virus shedding in the environment and of community transmission. The timeline illustrates the inherent delays in conventional WBE using centralised lab-based testing, which requires samples to be transported and processed at a high time, labour, and economic cost. By contrast, the near-source wastewater surveillance tool will enable rapid results in around 2 h with a 'Lab-in-a-suitcase' workflow that is amenable to low-resource field settings **b**-i. **b**-ii Sequences of the modified primers sets used for SARS-CoV-2 E-gene detection by dipstick. **b**-iii The process of near-source tests, from left to right, as: on-site sample extraction and preparation; one-pot RT-RPA; and advanced sensing nanomaterial assisted dipstick test for ultra-sensitive WBE.

interrupt transmission at an earlier stage has the potential for major human and economic benefits[19–22].

However, WBE is technically challenging[23]. Viral targets are heavily diluted in wastewater, preventing the detection of low abundant viruses, and the wastewater sample matrix is complex and variable, containing multiple inhibitors including humic substances and other compounds that can interfere with current testing methods[24,25]. This necessitates time-consuming clean-up and complex viral concentration steps[24,25] but can be overcome by a robust assay. The current gold-standard WBE method of PCR (digital droplet PCR, ddPCR, in the US, or reverse-transcription quantitative PCR, RT-qPCR) requires sample collection, cold-chain transportation to laboratory, and specialised equipment with staff for sample processing, concentration or virus cultivation (for polio), and performing PCR[26–30]. While conventional WBE requires sample transportation and centralised laboratory testing, near-source approaches offer significant advantages beyond just preserving nucleic acid integrity (Fig. 1a). These include dramatically reduced turnaround times ( > 24-72 h) for public health decision-making enabling more rapid interventions such as prophylactic treatments and ward closures[31], lower overall testing costs by eliminating transportation and laboratory infrastructure, and accessibility in locations with limited laboratory resources, such as wastewater pumping stations, schools, hospitals, airports, prisons, or care homes in developed and developing countries[32–34].

To facilitate this, point-of-care sensor technologies are required[35]. Mass population testing with lateral flow tests (LFTs) has been shown to be feasible and acceptable to non-specialist users, e.g., self-testing[31]. These tests typically use gold nanoparticles creating a visually readable test line, but lack the sensitivity required for WBE[36]. Several reports combine lateral flow tests with molecular nucleic acid amplification

assays such as loop-mediated isothermal amplification (LAMP) and CRISPR[37,38]. However, current paper-based sensors typically achieve gene copy numbers in the thousands per μL, falling short of the requisite sensitivity[37]. Additionally, these assays often involve high temperatures (65 °C for LAMP) and many steps, resulting in reporting delays and rendering them less suitable for near-source resource-limited settings. Few tests use Recombinase Polymerase Amplification (RPA), which works at lower temperatures of 37-42 °C, so is more amenable to resource limited settings[39,40]. The pairing of CRISPR with RPA might increase specificity, but CRISPR requires a relatively long testing time and adds cost and complexity. There remains an unmet need for suitable rapid tests amenable to near-source WBE.

The introduction of alternative nanoparticle probes for WBE has received comparably little attention[31,41]. A variety of nanomaterials including carbon black, magnetic, enzymatic, and fluorescent such as upconverting nanoparticles are emerging for use in LFTs with applications in WBE[42,43]. Of these, fluorescent nanomaterials show promise as a more sensitive label than colloidal carbon or gold, but the auto-fluorescence observed from both the sample matrix and from the nitrocellulose membrane that makes up the test strip limit the overall sensitivity. To overcome this, fluorescent nanodiamonds (FNDs) containing nitrogen-vacancy centres show promise in the emerging field of quantum sensors. Their spin-dependent emission intensity can be modulated using a microwave field[44]. They have been shown to improve detection limits by 100,000 with model biotin-avidin assays and 7500-fold over gold nanoparticles for amplicon detection, improving signal-to-noise ratio (SNR) on lateral flow test by background separation[45]. This provides an opportunity for sensitive, non-subjective test interpretation[41] and reporting, aligning with the REASSURED criteria for the development of an ideal rapid test, which specify real-time data capture and connectivity (in addition to ease of specimen collection, being affordable, sensitive, specific, user-friendly, rapid and robust, equipment free, and deliverable to end-users)[46].

Herein we report approaches in near-source WBE through the development of two complementary nanoparticles-based rapid lateral flow detection platforms, focusing on SARS-CoV-2 as an exemplar. Our platform includes: 1) the implementation of commercially available carbon black nanoparticles (CBNPs) for visual readout in WBE rapid testing, providing an accessible, equipment-free detection option; and 2) the application of spin-enhanced FNDs in a dipstick format for wastewater surveillance, achieving ultra-sensitivity in field-deployable pathogen detection. Recognising that near-source WBE requires streamlined workflows, we develop a user-friendly one-pot assay integrating lyophilised RPA, significantly reducing procedural complexity. We further simplify sample concentration and magnetic bead extraction methods, enabling rapid on-site sample processing. The integration of these technologies with a portable nanodiamond reader creates a complete lab-in-a-suitcase solution suitable for deployment at wastewater sources (Fig. 1b-i, iii). To validate this approach, we conduct a comprehensive blinded pilot study using 62 raw wastewater samples from the Welsh national WBE programme, demonstrating real-world applicability and performance.

## Results
### Primer design and RPA assay optimisation
The ability to detect fragments of SARS-CoV-2 in the complex matrix of wastewater is challenging, and relies on careful design of the RPA primers to target conserved regions of the genome with high sensitivity and specificity. Our RPA assay was designed to target multiple gene regions in a one-pot reaction. The gene target with higher performance then was chosen for the assay. Conserved regions of the SARS-CoV-2 genome in the envelope (E) gene and the RNA-dependent RNA polymerase (RdRp) gene were selected based on BLAST analysis of 100% identity of SARS-CoV-2 and low identity against other coronaviruses[47] (Fig. 1b-ii, Supplementary Table 1). Three pairs of RPA

primers were designed and evaluated to optimise performance on dipstick tests, covering test sensitivity, specificity, and time to result. Our approach builds on our previous work developing RPA assay for detection of SARS-CoV-2 in clinical nasal swabs. which resulted in high sensitivity and specificity against common seasonal coronaviruses, SARS-CoV-2 and MERS-CoV model samples[47]. We also developed a freeze-drying approach, whereby reagents were snap-frozen in liquid nitrogen and lyophilised (freeze-drying) for one-pot assay.

### RPA assay optimisation on commercial carbon black nanoparticle dipstick tests
The first assay uses RPA with carbon black nanoparticles. Although the exact composition of the test is proprietary, the particles used in the commercial assay by Abingdon were characterised by TEM and found to be typically 20-50 nm in diameter (Supplementary Fig. 1). The selected primers (~30 bp) were designed with biotin and digoxigenin (DIG) or carboxyfluorescein (FAM) functionalities at each end, enabling the amplicons (~200 bp) generated from the modified primers to bind to commercial dipstick with anti-DIG or anti-FAM antibodies deposited test lines on the strip on one end, and neutravidin functionalised CNP on the other end (Fig. 2a, Supplementary Fig. 2). To perform this rapid assay, preconcentrated wastewater samples (prepared through polyethylene glycol (PEG) precipitation[26]) were mixed with RPA reagents, and incubated at 37 °C, to amplify the target genes. This low-cost dipstick strip (around £2 per strip based on current academic low-volume purchasing at the time of the study) was dipped directly in the analyte to generate a visually detectable grey test line due to the accumulation of carbon black particles, which can be read by the naked eye or a smartphone camera[47] (Fig. 2a-ii, iii). The performance of the RPA-CBNP assay was optimised via screening of primer concentration, assay temperature, and magnesium ion buffer concentrations. To determine the assay time-to-result, the signal was measured at different time points of the reaction ranging from 15 to 40 min. The ideal assay time was chosen as 40 min yielding the highest signals on dipstick strips without non-specific amplification (Supplementary Fig. 3a). The longer amplification generates significant levels of nonspecific amplification. To further investigate the robustness of the assay, we tested humic and fulvic substances, which are commonly found in wastewater and known to interfere with other amplification approaches, but found that our RPA reaction was not significantly affected on low concentration of contaminants[48] ($P < 0.001$) (Supplementary Fig. 3b).

Next, the limit of detection (LoD) of the RPA-dipstick-CBNP assay was investigated using model WBE standard samples, consisting of a serial dilution of synthetic whole SARS-CoV-2 omicron RNA genome spiked into SARS-CoV-2 negative wastewater received from the Welsh WBE programme. The RPA assay was performed against the dilutions for 40 min, and the amplification products were run on CBNP dipsticks. The LoDs were found to be 71 (95% Confidence Interval (CI): 34–134) gene copies per reaction for the E gene and 127 (95% CI: 64–249) gene copies per reaction for the RdRp gene (Fig. 2c, Supplementary Fig. 3c). 62 wastewater samples were evaluated by this assay. To ensure direct comparability between CBNP and FND detection methods, all 62 wastewater samples were processed using this identical magnetic bead extraction protocol (will be detailed discussed later)[26]. 41 of these samples were RT-qPCR confirmed positive (with cycle threshold ($C_t$) ranging from 25 to 40) and 21 were considered negative for SARS-CoV-2 ($C_t > 40$). All samples were assessed on the RPA assay, showing semiquantitative results when test line intensity is compared against $C_T$ values. Receiver operating characteristic (ROC) curve analysis indicates the 80% sensitivity and 100% specificity of the RPA-dipstick-CNP assay when applied on samples of $C_t$ ranging from 25 to 40 (Fig. 2d). This corresponds to 8 false negatives and an inspection of Fig. 2e indicates these are at high $C_t$ values of 35–40, correlating to samples with very low virus concentrations. The results reveal that

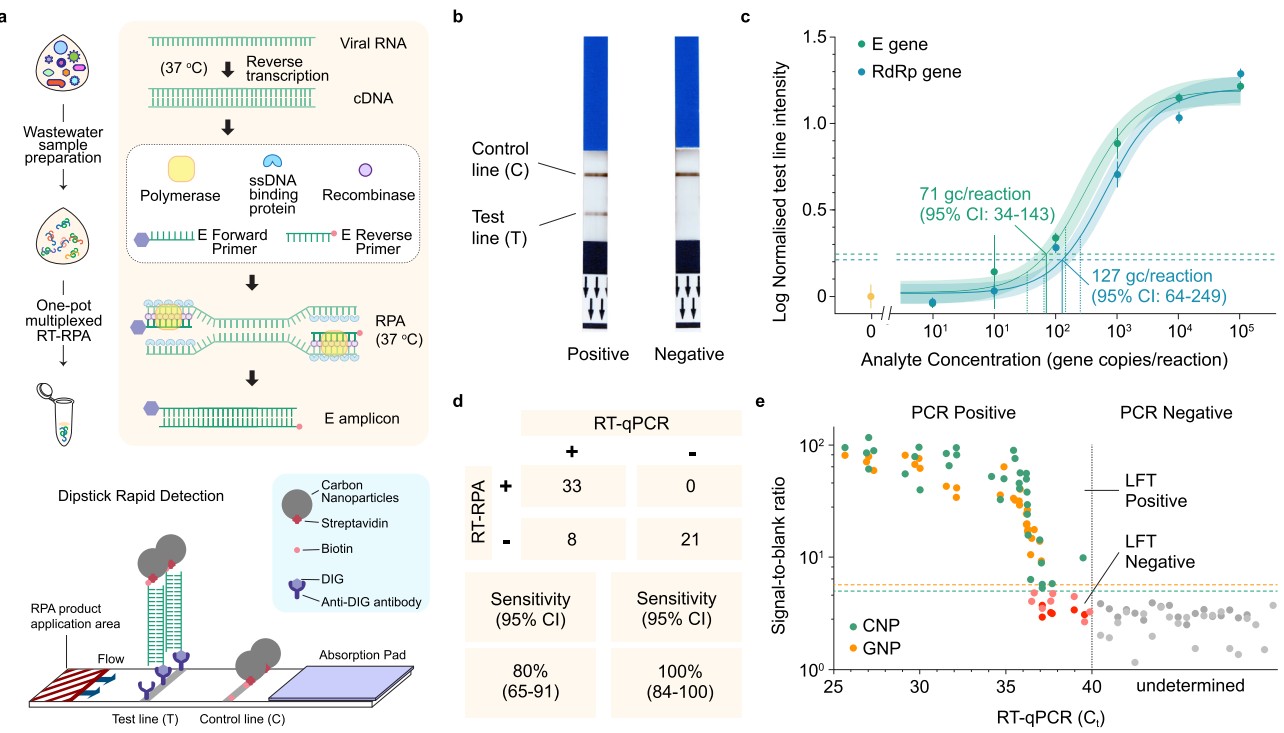

**Fig. 2 | Development of RPA-dipstick assay with carbon black nanoparticles (RPA-dipstick-CBNP) for rapid SARS-CoV-2 fragments detection in wastewater.** **a** Schematic of RPA-dipstick assay for wastewater SARS-CoV-2 surveillance. RNA extracted from wastewater was amplified to generate E gene amplicons, which can bind to a complementary test line on the dipstick strip. The use of carbon black nanoparticles as sensing probe allows test results to be visualized by naked eye or through camera capture. **b** Examples of test strips showing a typical positive and negative result. **c** Analysis of the RPA-dipstick-CBNP assays show LoDs of 71 gc/reaction (95% CI: 34–134) for the E gene assay (green) and 127 gc/reaction (95% CI: 64-249) for RdRp gene assay (blue). The E-gene test was chosen for the further development. The dots show means and error bars show the s.d. of repeat measurements ($n = 3$ technical replicates and $n = 3$ measurement replicates for each sample). A stretched exponential regressions are shown by solid lines, and shaded areas show the 95% confidence intervals of the fits. **d** The sensitivity of the assay was 80% (95% CI: 65-91) and the specificity was 100% (95% CI: 84-100) ($n = 62$). **e** The semiquantitative nature of the assay has been demonstrated by plotting wastewater sample RT-qPCR cycle threshold ($C_t$) against the RPA-dipstick assay (with carbon black or gold nanoparticles probe) test line intensities, showing dose-dependent results. RT-qPCR–positive wastewater samples (with $C_T$ range from 25 to 40) and dipstick tested positive samples are plotted in green dots (CNP) or yellow dots (GNP), and RT-qPCR–negative samples ($C_t > 40$) and dipstick tested negative samples are plotted in pink dots (CNP) and red dots (GNP). PCR undetermined samples are plotted in grey dots. Significant differences were determined using one-way analysis of variance (ANOVA) with Tukey's post hoc test. $P = 0.2007$ between CNP and GNP groups. The assay is semi-quantitative due to saturation in the regime where there are already many amplicons per nanoparticle (so there is a marginal increase in nanoparticle-test line binding rate with increasing amplicon concentration). Source data are provided as a Source Data file.

although the CBNP assay can detect multiple genes of SARS-CoV-2, the assay still needs to be improved for detecting ultra-low copies of targets ($C_t > 35$) that can be associated with the early stages of an outbreak. A comparison between CBNP and conventional gold nanoparticles (GNP) with same LFT setup presented the robust performance of the assay (Fig. 3e).

**Spin-enhanced fluorescent nanodiamond dipstick tests for WBE**
To further enhance WBE assay sensitivity, we used spin-enhanced nanodiamond dipsticks. Our approach uses the high brightness of fluorescent nanodiamonds, which arises from nitrogen-vacancy defects (3 ppm), together with the ability to modulate their fluorescence emission by manipulating their spin state with an external microwave field. This allows the removal of background autofluorescence from the nitrocellulose strip, which typically limits sensitivity. This approach is in principle suitable for complex samples as it is inherently background removing, especially for WBE. Firstly, the FNDs were functionalised with anti-DIG antibody. In the presence of the amplicon, the FND binds to the test line by creating a sandwich between the DIG labelled tail of the amplicon, the particle, and the

polystreptavidin deposited test line[39] (Fig. 3a). FNDs (600 nm diameter) were characterised by Scanning Electron Microscopy and Dynamic Light Scattering showing in Fig. 3b. Using the primers and conditions developed for carbon black particles above, we then tested different amplification times on an FND-based assay. A benchtop microscope setup was used to detect the presence of the FND particles, with a resonator integrated on the sample stage (Fig. 3a)[25]. The microwave signal modulation approach enables lock-in detection with a lower LoD than imaging the test line (Fig. 3c, Supplementary Fig. 4). We found that only 25 mins RPA was required (compared to 40 mins with CBNP) marking a 37.5% reduction in assay time. The assay readout sensitivity improvement of 15375-fold over CBNPs to FNDs means that we can amplify for just 25 min to achieve higher enough sensitivity using FND dipsticks (Supplementary Fig. 5, and see Methods). The LoD for the RPA-dipstick-FND assay for E gene was found to be 7 (95% CI: 3–13) gene copies per reaction (Fig. 3d), compared to 71 copies for CBNPs, giving a ten-fold improvement in half the time. We then evaluated real wastewater samples and compared the signal to blank ratio for FNDs against the RT-qPCR cycle threshold ($C_t$) (Fig. 3f) to find that the sensitivity of the assay was 100% (95% CI: 91-100) and the specificity

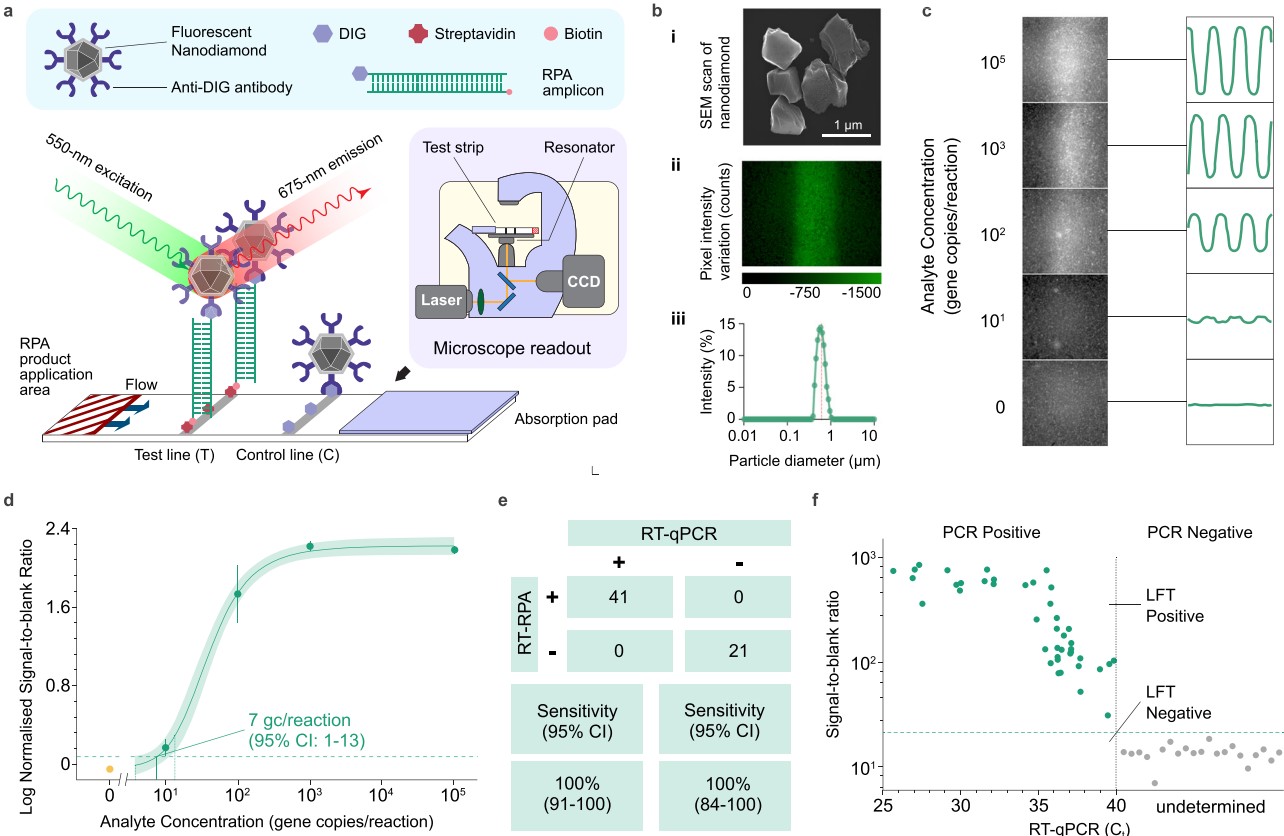

**Fig. 3 | Development of RPA-dipstick assay with fluorescent nanodiamond particles (RPA-dipstick-FND) for rapid SARS-CoV-2 detection in wastewater.** **a** Schematic of RPA-dipstick-FND assay. The FND was modified with anti-DIG anti-bodies, which bind to the DIG labelled amplicon that, in-turn, binds with the test line. The test strips can be imaged by microscope with an amplitude-modulated microwave field. **b** Characterisation of FNDs used for dipstick. i Scanning electron microscope images of FNDs with particle core diameters of 600 nm. ii The pixel variation of the test line (with immobilized FNDs) was shown as the microscope imaging of the assay. Scale bar: 1 μm. iii Nanoparticle size analysis. DLS measured the size and aggregated fraction after functionalization of antibody, displaying peak particle hydrodynamic diameters of 600 nm. $n = 3$ measurement replicates for each sample. **c** Comparison between lock-in and conventional fluorescence images analysis. Captures of measured RPA-dipstick-FND test lines (left) from selected concentrations of a serial dilution, with the corresponding fluorescent intensity time series plots (right), demonstrating that signal modulation enables

detection well below the concentration required to directly image the test line (see extended plots in Supplementary Fig. 4). **d** Limits of detection for RPA-dipstick-FND assay for E gene were 7 gene copies per reaction (95% CI: 3–13). The dots show means and error bars show the s.d. of repeat measurements ($n = 3$ technical replicates and $n = 3$ measurement replicates for each sample). A stretched exponential regressions are shown by solid lines, and shaded areas show the 95% confidence intervals of the fits. **e** The results reveal that the assay picked up on all positive samples. The sensitivity of the assay was 100% (95% CI: 91–100) and the specificity was 100% (95% CI: 84-100) ($n = 62$). **f** The wastewater sample RT-qPCR cycle threshold ($C_t$) plotted against RPA-dipstick assay test line intensities (E gene) demonstrates dose-dependent semiquantitative results. RT-qPCR–positive waste-water samples (with $C_t$ range from 25 to 40) and dipstick tested positive samples are plotted in green dots, and RT-qPCR–negative samples ($C_t > 40$) and dipstick tested negative samples are plotted in red. Source data are provided as a Source Data file.

was 100% (95% CI: 84–100) (Fig. 3e), marking a significant improvement on the CBNP dipstick sensitivity. The cross-comparison of sensitivity and specificity between different probing nanoparticles indicates a big improvement using FND assisted dipsticks (Supplementary Fig. 6).

**Towards a near-source nanodiamond WBE assay**

Building on the excellent performance of nanodiamond assay using a benchtop microscope described above, we then moved towards developing a proof-of-concept lab-in-a-suitcase prototype for rapid near-source testing of raw, unprocessed wastewater samples. Our approach combines on-site sample preparation, amplification, and readout. Sample preparation is carried out through syringe filter-trapping and concentration, coupled with a magnetic bead sample enrichment and viral RNA extraction step. Freeze-dried RPA pellets reduce the complexity of the one-pot RT-RPA reaction, and nanodia-mond based dipsticks are read with a small portable fluorescence reader (customised device to excite FNDs at 550 nm and readout at 675 nm, Axxin) for near-source ultra-sensitive results (Fig. 4a,

Supplementary Fig. 6a). All these small portable components can currently be used in a lab-in-a-suitcase format, forming wastewater epidemiology quantum on-site sensing system (WE-QOSS). This allows rapid field testing at the site of an outbreak, such as a care home, quarantining facility or pumping station, with real-time connectivity to transfer results to surveillance systems. Moreover, the FND user-friendly reader offers a simple user interface, touch screen navigation and connectivity of results over LAN or Wi-Fi and is compatible with Laboratory Information System and other health databases.

The complexity of wastewater samples necessitates sample fil-tration and extraction. To overcome the complexity of the time-consuming wastewater concentration methods used in wastewater monitoring programmes[49], we began to explore a much simpler syr-inge filtration and in situ nucleic acid extraction method which had shown promise concentrating and isolating viral RNA from wastewater samples on site and to test the compatibility with our nanodiamond assay. By streamlining the sample preparation and assay timeline, we aim to reduce the overall turnaround time from sample collection to result to approximately two hours – marking a significant advance on

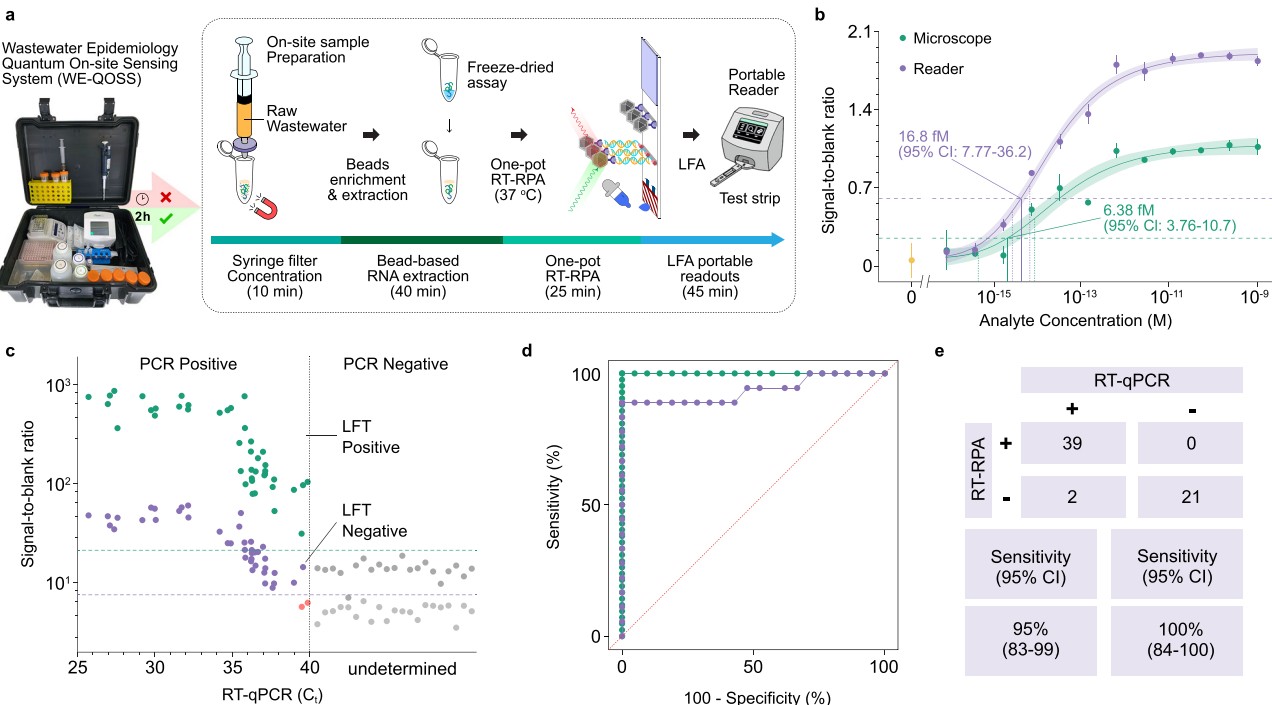

**Fig. 4 | Proof of concept and evaluation of for SARS-CoV-2. a** Setup of the lab-in-a-suitcase platform for two-hour near-source WBE testing. The four stages of the assay and their timelines are shown, including syringe filter sample enrichment (10 min), magnetic beads RNA extraction (40 min), pre-lyophilised one-pot RT-RPA reaction (25 min), and nanodiamond assisted dipstick test with a portable reader readout (45 min). **b** LoD comparison of two readouts (microscope and reader) based on the model amplicon binding FND strip tests. The microscope readout LoD was 6.4 fM (95% CI: 3.8–11), and the portable reader readout LoD was 17 fM (95% CI: 7.8–36), making the latter 2.6-fold less sensitive in a model system. The dots show means and error bars show the s.d. of repeat measurements ($n = 3$ technical replicates and $n = 3$ measurement replicates for each sample). A stretched exponential regressions are shown by solid lines, and shaded areas show the 95% confidence intervals of the fits. **c** The assay was evaluated on 62 raw wastewater samples, treated via magnetic beads extraction. The wastewater sample RT-qPCR cycle threshold ($C_t$) was plotted against nanodiamond signals generated both from the microscope readout and from the portable reader. **d** The receiver operating characteristic curve analysis of the evaluation results on the raw wastewater samples ($n = 62$). **e** The sensitivity of the WE-QOSS platform is 95% (95% CI: 83–99) (which compare to the 100% (95% CI: 91–100) based on microscopy readout), and the specificity is 100% (95% CI: 84–100). Green: microscope, Purple: reader. Source data are provided as a Source Data file.

the usual 24-to-72-hour reporting lag associated with gold-standard laboratory based qRT-PCR. A breakdown of the steps involved is shown in Fig. 4a. 10 min are required for a 50 mL wastewater sample to be syringe filtered (0.45 μm pore size membrane to remove large particulates) and chemically lysed; 40 mins is required for magnetic bead extraction of RNA fragments, followed by 25 min of the one-pot RT-RPA assay and 45 min for the FND-based dipstick test using a portable reader readout (Axxin, non-modulation detection only) (Fig. 4a). The assay has been adapted to make it amenable to the reader by changing the nanodiamond concentration (see methods). The model amplicon comparison of dipstick is shown in Fig. 4b, finding an LoD of 6.4 fM (95% CI: 3.8–11) on the benchtop microscope, and 17 fM (95% CI: 7.8–36) on the portable reader. While the benchtop microscope has higher sensitivity, this comes at a higher cost (£30 K + ) and lacks portability. Whereas the smaller footprint, weight, and cost of the portable reader (measuring just 12x11x12 cm, weighing 650 g, and a list price of $8 K, Axxin) brings significant advantages for near-source field testing, overcoming the need to transport samples to centralised labs without compromising on sensitivity and specificity (Supplementary Table 2). Our workflow processes 50 mL of raw wastewater down to 50 μL of extracted RNA, with the FND method's detection limit of 7 copies per reaction (95% CI: 3–13 copies), based on the standardised RNA samples, translating to approximately 140 genome copies per litre, gc/L (95% CI: 60–260 gc/L), in the original wastewater sample, assuming ideal RNA recovery efficiency. This sensitivity falls within the lower range of SARS-CoV-2 RNA concentrations typically reported in WBE studies ($10^2$–$10^5$ gc/L) during the pandemic[50], making it suitable

for early outbreak detection. Notably, this detection capability is comparable to established laboratory qPCR methods used in national WBE surveillance programmes, which achieve limits of detection around 611 gc/L when processed through conventional concentration steps with similar RNA recovery efficiency[51] (see Methods and Supplementary Fig. 7c). The ability to achieve this sensitivity level with a field-deployable platform represents a significant advancement for near-source WBE applications. Next, we tested 62 untreated wastewater samples from the Welsh national WBE programme on the portable sensing platform. Figure 4b shows the comparison of the wastewater sample RT-qPCR cycle threshold ($C_t$) plotted against nanodiamond signals generated from the lab-in-a-suitcase assay, comparing the microscope and portable reader readouts. The microscope readout with microwave modulation could detect all 41 SARS-CoV-2 positive raw wastewater samples. The portable reader (with no modulation) performed well and detected 39 out of 41 samples, only missing 2 samples in the ultra-low viral concentration range ($C_t$ 39–40) (Fig. 4c, Supplementary Fig. 6). The receiver operating characteristic curve (ROC) analysis indicated the sensitivity of the lab-in-a-suitcase assay being 95% (95% CI: 83–99) with a specificity of 100% (95% CI: 84–100) (Fig. 4d, e). The evaluation indicates the capability of using our portable quantum sensing platform in near-source wastewater pathogenic sensing. While our assay demonstrates a concentration-dependent response, it is important to note that it is primarily semi-quantitative in nature (same applied to CNP assay and FND-microscopy assay). The stronger linearity observed in the middle concentration ranges is sufficient for many WBE applications,

particularly for early outbreak detection where binary positive/negative results can trigger appropriate investigation protocols. The inherent variability in wastewater matrices - including fluctuating individual shedding rates, dilution effects, and RNA degradation - can contribute to the non-linear response observed at very low and very high viral concentrations. Our experience with national WBE programmes suggests that this level of quantification is adequate for near-source applications focused on early warning capabilities. Future refinements could enhance quantitative performance through improved sample concentration methods (e.g., increased sample volumes or portable ultrafiltration) and advanced binding kinetics analysis, building on our previous work[52]. As a result, the lab-in-a-suitcase format enables complete on-site processing - from sample collection to result interpretation - without requiring sample transportation or specialised laboratory facilities. This approach addresses several key limitations of conventional WBE in turnaround time and high costs and extends WBE capabilities to underserved or remote locations. While amplified DNA stability is not a significant concern, the operational advantages of integrated near-source testing justify this approach for rapid, accessible wastewater surveillance.

## Discussion

In summary, we report an important solution in the field of near-source WBE, with the development of two complementary near-source tests, with the potential to open up a field of near-source WBE surveillance with significantly faster turnaround times, providing early warnings of outbreaks: 1) a low-cost carbon black particle dipstick assay which can be read by eye; 2) an ultrasensitive and field portable quantum fluorescent nanodiamond assay requiring only 2 h from raw WBE sample collection to results read-out. The main advantage of our approach and workflow is that the sample process can be performed not only at wastewater treatment plants but also in near-source settings such as prisons, hospitals, and other health care facilities (or within the sewer network at manholes/pumping stations etc) and also in resource limited settings in low and middle income countries, where PCR costs might be prohibitive. In these settings, early (presymptomatic) detection of cases using WBE approaches would be crucial in reducing the number of cases and preventing nosocomial outbreaks, which often necessitate ward closures and delays in treatment, potentially resulting in loss of life and additional economic burdens. The workflow proposed for our nanodiamond enhanced dipstick would readily fit inside a small van, enabling mobile and proactive monitoring - something that would not be feasible with a real-time PCR machine, which is sensitive to vibrations and agitation. We acknowledge that some approaches for on-site testing at wastewater treatment plants do exist (e.g. Cepheid® LuminUltra®), however, these remain significantly more expensive (£10,000-20,000, Supplementary Table 2) than our reader and our approach is designed to be much more field-portable, low cost and deployable in a wider range of use-cases (see below). Moreover, we point out that the level of training needed for PCR is generally considerably higher than for a lateral flow or dipstick test.

Our findings suggest that carbon black nanoparticles are comparable to traditional gold particles, yielding 80 and 82% sensitivity respectively and 100% specificity. Moreover, carbon and gold particles required a longer 40-minute amplification step. By contrast nanodiamonds achieved 100% sensitivity and 100% specificity with a benchtop microscope readout with just 25 min RPA amplification. The FND readout was demonstrated a 15,375-fold improvement in sensitivity compared to CBNP when tested with identical ssDNA model amplicon on LFTs. Moreover, a portable nanodiamond assay achieved 95% sensitivity and 100% specificity with a 25-minute RPA amplification, even without modulation. We streamlined the sample preparation process using syringe capture enrichment, magnetic beads extraction, and lyophilisation of the RT-RPA reaction, minimising user intervention and maximising portability to render it particularly well-suited for near-source wastewater surveillance. As a result, this platform stands as a saving of public resource, and is especially well-suited for resource-limited or decentralised settings, including remote hospitals, refugee/transit camps, care homes, and prisons, in both developed and developing countries. This is a significant advance on current gold-standard laboratory tests, not only drastically reducing testing time (from an estimated 72 h with conventional centralized lab RT-qPCR tests to just 2 h), but also substantially cutting costs.

Our adaptable platform technology is not limited to SARS-CoV-2 but has the capacity to detect a wide spectrum of nucleic acid fragments from the various pathogens present in wastewater and is amenable to highly multiplexed formats[53]. In future, we aim to field-test the 'lab-in-a-suitcase' concept with diverse end-users, developing Standardised Operating Procedures (SOP), that include quality control protocols such as preserving test strips for laboratory verification or performing replicate testing to ensure reproducibility. In future such SOPs could incorporate standardised freeze-dried reagent kits with built-in controls to minimise variability across testing sites, while also exploring the integration of readers, modulation and smartphones with data connectivity for real-time geospatial data collection, automated analysis and potential remote quality control oversight. In addition, we also noticed that SARS-CoV-2 is typically present at higher concentrations in wastewater compared to pathogens such as polio[54]. Future developments would focus on enhancing RNA recovery efficiency from complex wastewater matrices - a critical factor for low-abundance targets. As demonstrated in our previous work[49], a straightforward approach involves increasing sample volumes from 50 mL to 100-150 mL, which significantly improves detection of rare targets. This modification, combined with the high sensitivity of our FND detection platform, could extend the utility of our approach to a broader range of pathogens with varying abundance levels in wastewater, while maintaining the advantages of rapid, field-deployable testing. If successful, this quantum-based sensing platform can improve near-source wastewater and environmental testing. This could encompass applications in regions where rivers and estuaries are susceptible to contamination, enabling biodiversity monitoring and offering substantial benefits in terms of human, animal, environmental, and economic well-being. Through these collective efforts, we aspire to advance the field of low-cost WBE, enhancing the capacity to detect and monitor viral contaminants in near-source environments, ultimately contributing to much faster outbreak response times and, consequently, improved public health outcomes.

## Methods
### Wastewater samples
Refrigerated autosamplers were used to obtain 24-hour composite wastewater samples (untreated throughout collection) on a daily basis from treatment facilities across Wales. All sample processing and concentration procedures took place within a Containment Level 2 (CL2)/Biosafety Level 2 (BSL2) facility in accordance with WHO protocols and national biosafety standards. To reduce contamination risk during sample preparations, personnel wore face masks and/or protective visors. Following testing, samples stored at -80 °C to enable future retrospective studies. Sample provision occurred under agreement with the Welsh Government. The RPA assay protocol involved filtering 50 ml sample aliquots through 0.45 μm pore-sized membranes using 60-ml syringes, followed by chemical lysis and subsequent magnetic extraction before RPA amplification.

### Multiplex RPA assay
Primer sequences targeting E and RdRP genes with modification (Supplementary Table 1) enable duplex detection using dipstick method featuring neutravidin-conjugated carbon black nanoparticles. E gene primer modifications included biotin and digoxigenin labelling for recognition at test line 1, while RdRP gene primers received FAM

and biotin modifications for detection at test line 2. Non-specific binding assessment due to primer dimer formation involved template-free control reactions (NTC) using modified primers across concentrations of 10 μM, 2 μM, 1 μM, and 0.5 μM. The optimised 50 μL reaction mixture comprised TwistAmp® basic RPA pellets reconstituted in 29.5 μL Rehydration Buffer (TwistDX), forward primers and reverse primers (2.1 μL each at 1 μM concentration), RNA template (1 μL for each target gene), reverse transcriptase (2.5 μL at 200 U/μL), and nuclease-free water (5.1 μL). Reaction initiation occurred through addition of 2.5 μL magnesium acetate (280 mM concentration). Incubation proceeded at 37 °C for 25 min on shaking (250 RPM) dry bath. Parameter optimisation encompassed multiple variables. Time-to-result optimisation involved dipstick performance evaluation across incubation periods of 15, 20, 25, 30, 35, 40, and 45 min.

## CBNP dipstick test

Following RPA amplification, 10 μL of the reaction product was combined with 140 μL of running buffer in a microplate well, into which the dipstick was subsequently immersed. Results were interpreted after a 10-minute development period. CBNP strip was imaged by digital camera, with subsequent analysis performed via Matlab software (version R2022a).

## PEG wastewater precipitation

Viral concentration from wastewater samples and subsequent viral RNA extraction utilised polyethylene glycol precipitation methodology. The protocol requires an initial sample volume of 100–200 mL of raw wastewater, ultimately concentrated to a final volume of 0.1 mL. The concentration process begins with centrifugation to remove large debris and particulate materials from the sample. Following pH optimisation, the clarified supernatant undergoes incubation with PEG8000/NaCl solution, after which virus-PEG complexes are recovered through centrifugation as a pellet. The recovered pellet undergoes resuspension, followed by viral RNA extraction and quantification via a single-step RT-RPA assay.

## Preparation of functionalised FNDs

Polyglycerol (PG)-coated 600 nm FNDs (custom purchase from Adámas Nanotechnologies, 1 mg/mL in DI water, ~3ppm NV centres) were conjugated to antibodies using disuccinimidyl carbonate (DSC). The desired volume of 600 nm FND-PG particles at 1 mg/mL (Adámas Nanotechnologies) were added to a protein low-bind tube and placed in the bath sonicator at high power for 5 mins. The particles are then centrifuged at 21130 g for 4.5 mins. The supernatant is removed and the pellet are resuspended in anhydrous N,N-dimethlyformamide (DMF, Sigma-Aldrich). The suspension is mixed well and sonicated for 2 mins at 100% power. The centrifugation and wash steps are repeated to remove water. After the last centrifugation step, the particles are resuspended in equal volume of 50 mg/mL DSC in DMF and placed on the Thermoshaker for 3.5 hrs at 300 RPM and 25 °C. Next, excess reagents are removed by a centrifugation and wash step in DMF. After the last centrifugation, the particles are resuspended in DI water. The anti-DIG antibody (Abcam) is quickly added to the activated FNDs at 2.74 μg per 100 μL and placed on the thermoshaker for 15-17 hrs at 250 RPM and 25 °C. Subsequently, 1 M Tris-HCl (pH 7.5, Thermo Fisher Scientific) is added to the solution to quench the remaining succinimidyl carbonates and left on the thermoshaker for 30 mins. The particles are centrifuged at 21130 g and washed with DI water to remove unbound reagents; this step is repeated three times. The final resuspension is in 0.1% wt BSA in PBS.

The functionalised FND concentration was measured by fluorescent intensity. A 2-fold dilution series of the stock 600 nm PG-FNDs and a 1:50 dilution of the functionalised FNDs were added to a black 96-well plate. The fluorescent intensity was measured on the

spectrophotometer (SpectraMax i3, Molecular Devices LLC). Full analysis details can refer to the Miller et al.[45].

## Dipstick assay based on FNDs

The FND-PG-anti-DIG preparation was diluted to a concentration of ~0.6 pM in 1X PBS. 5 μL of the FND dilution was mixed with 65 μL of running buffer (5% milk + 0.05% Empigen in $H_2O$) in a 96-well plate. Subsequently, 5 μL of the post-RPA product was added to each well and allowed to bind for 10 mins. The half-strip LFA with polystreptavidin test line (Mologic) was added to the well and allowed to run for 15 mins. The strips were then washed with 75 μL of wash buffer (1X PBS + 0.5% Tween20) and allowed to dry before test line read-out.

## Fluorescence measurements and analysis

Optically detected magnetic resonance (ODMR) measurements were carried out on modified fluorescent nanodiamonds. 600 nm FND-PG-BSA-biotin was used to run on a polystreptavidin LF strip at a high concentration (1:5 dilution) to generate a strong positive signal test line. The strip was placed directly on the resonator board fixed to the microscope stage with microwaves being supplied using a (SynthUSB3) source and microwave amplifier (ZRL-3500 + , Mini Circuits) through a linear strip line to sweep across frequencies from 1 to 3 GHz. The distance from resonator to test line is approximately 1 mm (the thickness of the test strip including backing card). A MATLAB code was used to plot the frequency intensity across different frequencies. (see ODMR measurements in Supplementary Fig. 8). The dipstick strips were imaged using a fluorescence microscope (Olympus BX51) with a 550 nm green LED excitation light source (CoolLED pE-4000) producing 60 mW at sample, a filter cube with an excitation filter (Semrock-500nm bandpass, 49 nm bandwidth), a dichroic mirror (Semrock-596nm edge), and 593 nm long-pass emission filter (Semrock). A 20x/ 0.4 BD objective lens producing a 2.1 mm beam diameter at focus was used. Images were recorded using a high-speed camera (Hamamatsu, ORCA-Flash4.0 V3) and HCImage Live software (Hamamatsu) for 15 s at a sample rate of 33.33 frames/sec, where the mean of each frame were calculated to give a time-series of mean pixel values. Typical photon count rates are $3.03 \times 10^5$ - $3.44 \times 10^5$ counts/s for weak to strong positive testing conditions (approximately 12% of the total photon count $\approx 4.04 \times 10^4$ counts/s contributed from FNDs). A voltage-controlled oscillator (VCO) (Mini-Circuits-ZX95-3360 + ) and a low-power amplifier (Mini-Circuits-ZX60-33LN + ) were connected to an omega-shaped resonator and circuit board (Minitron, Rogers 4003c 0.8 mm substrate with 300 gm$^{-2}$ copper weight) to generate the microwave field (+ 17 dBm input power at 2.87 GHz). The modulated signal is achieved by modulating the VCO input with a reference frequency generator at 4 Hz, using a 32.768 Hz crystal oscillator (Farnell, DS32KHZ) and 14-stage frequency divider (Farnell, CD4060BM). The microwave circuit and full microscope set up described in Miller et al.[45].

## FND assay limit of detection analysis

A statistical method for determining the LoD was employed based on previously reported methods reported[55]. The LoD is defined as the intersection of upper 95% confidence interval of the negative controls with the lower 95% confidence interval of the lowest detectable concentration (a sample exactly at the LoD). The latter confidence interval was calculated using a characteristic variance for positive samples: the average pooled variance of the positive samples. To facilitate this, a log transform of the concentrations, $C$, and lock-in signals, $y$, was used to normalise variances, allowing all $y$-values to be used to calculate the variance, not just those close to the LoD:

$$y^* = \log_{10}(y + 2) \tag{1}$$

$$C^* = \log_{10}(C + 2) \tag{2}$$

See Miller et al.[55] and Holstein et al.[56] for further details. The data were then fitted to a stretched exponential model:

$$y(C) = a + b\left(1 - e^{-\left(\frac{C}{k}\right)^d}\right) \tag{3}$$

$$y^*(C^*) = \log_{10}\left[a + b\left(1 - e^{-\left(\frac{10C^*-2}{k}\right)^d}\right)\right] \tag{4}$$

where $a$, $b$, $k$, and $d$ are fitted parameters. This model was selected as the best fit based on the root mean squared errors (RMSE) and the small-sample corrected Akaike Information Criterion (AICc) parameters, as described in Miller et al.[55]. The fitting software is open source and available at GitHub (https://github.com/bensmiller/detection-limit-fitting/). The confidence interval of the LoD is defined as in Holstein et al.[56].

### Magnetic Beads wastewater extraction

The whole process contained in syringe wastewater sample filtration concentration, and RNA extraction with magnetic beads, followed by isothermal amplification, and dipsticks for amplicon detection. The wastewater samples directly collected from sampling points were well mixed. Each 50 mL wastewater sample was passed through a syringe filter (Millex™ Nylon syringe filter, 0.45 μm) with a 60 mL syringe. Coarse material with target organisms was concentrated by the filter. After the addition of 400 μL of lysis buffer in the syringe, the filtered water sample was incubated at room temperature for 10 min for RNA release. The released RNA (400 μL) was eluted into a PCR tube with the syringe and mixed with MagaZorb® (Promega) magnetic beads extraction reagents for RNA extraction and purification. Finally, 50 μL of extracted RNA was eluted, and amplified in an RPA reaction. The RPA reaction was incubated within the thermos with at 37 °C, for 25 min or 40 min, for FND readouts or CNP readouts respectively. The amplicon was detected with the dipstick tests. The amplification time was optimised differently for CBNP and FND detection methods based on their sensitivity differences. As demonstrated in Supplementary Fig. 5, FND detection is approximately 15,375-fold more sensitive than CBNP when using identical amplicon concentrations. This substantial sensitivity enhancement allows the FND method to reliably detect amplicons after just 25 min of RPA, while the CBNP method requires 40 min to achieve adequate signal strength. This optimization balances detection sensitivity with the need for rapid turnaround in near-source applications. Detection sensitivity in WBE applications relies on effective sample concentration and RNA extraction protocols that maximise recovery from original wastewater samples. To assess extraction efficiency, we conduct a comparative analysis between our magnetic bead method and conventional PEG precipitation across 30 raw wastewater samples. Both methods were evaluated using identical qPCR primers and conditions. The magnetic bead extraction demonstrated comparable recovery efficiency (two-tailed $t$-test $P = 0.2254$, $t = 1.239$), shown in Supplementary Fig. 7c. This comparison confirms that our streamlined magnetic bead extraction provides suitable recovery rates for reliable near-source detection while significantly reducing processing complexity and time.

### Lyophilisation of RPA reactions

The multiplex one-pot RPA assay was prepared as described above in RPA assay section, however without MgOAc. The tubes were opened, sealed with parafilm and small perforations were made using a needle. The tubes were snap-frozen in liquid nitrogen and lyophilised (freeze-drying) for 4 h (temperature -50 °C, vacuum set point 0.025 mbar) (see Supplementary Fig. 7a, b). The lyophilised reactions were kept in a -20 °C freezer for long term storage. The heat stability of the lyophilised reactions at room temperature (20 °C) was revealed to be longer than one month.

### Portable reader evaluation

The commercially available portable fluorescent reader (AX-2X-S Axxin) designed to provide quantitative or qualitative results for visible or fluorescent immunoassays, was adapted to specifically detect FNDs on dipsticks. The portable fluorescence reader was set-up using the corresponding software to set test parameters including test line position. The reader includes an adjustable carrier for dipstick strips and a GUI for running stand-alone tests, as well as connectivity of results over LAN or Wi-Fi and Laboratory Information System Compatible. The fluorescent reader scans along the specified length of the dipstick strip and detects a peak at the test line position. The peak is evaluated using the peak height intensity (mV). The sensitivity of the reader was evaluated using model amplicon to compare the microscope values with and without modulation (microwave modulation of FND emission applied in our previous work).

### Statistics & reproducibility

Fluorescence microscope (Olympus BX51, CoolLED pE-4000) with Matlab 2019a and HCImageLive 4.4.1.0 were used to collect data for the microscope readout of this study. Portable fluorescent reader (Axxin AX-2X-S) with Kinetic Designer Software V2.0 were used to collect data for the portable readout of this study. Data analysis was performed in Matlab 2019a and R2022a. The fitting software is open source and available at GitHub (https://github.com/bensmiller/detection-limit-fitting/). GraphPad Prism V10.3.0 was used to generate plots and kinetic curves. As for the part of proof-of-concept study, determining analytical sensitivities, as opposed to clinical sensitivities, small sample sizes are suitable and no sample size calculation was necessary. 62 raw wastewater samples from Welsh National WBE programme were evaluated in this study. Details of the number of replicates for each figure are given in the text and figure captions. No data were excluded from analysis. Reproducibility was ensured by measurements from multiple technical replicates. The exact number of independent technical experiments is mentioned in the figure legends. No results are included that were not observed in multiple experiments. All attempts at replication were successful. There was no randomisation as the assay development reply on the model samples from serial dilutions of target (synthetic RNA), so there were no covariates. The wastewater evaluation were designed to compare the diagnostic performance of the conventional pathogen identification. All samples were equally applied to both methods. Samples were randomised upon giving them to the study staff, who were kept blinded to the SARS-CoV-2 RT-PCR results of the samples when they performed validation experiments for the assay developed in this study.

### Reporting summary

Further information on research design is available in the Nature Portfolio Reporting Summary linked to this article.

## Data availability

All data generated or analysed during this study are available within the Article, its Supplementary Information file, and the Source data file. The full image dataset is available from the corresponding author on request, in line with UCL and funder's requirements (*EPSRC* and *UKHSA* policy framework on research data). We will aim to process all requests within two weeks. Source data are provided with this paper.

## Code availability

The computer code used during the current study are available from the corresponding author on request, in line with UCL and funder's requirements (*EPSRC* and *UKHSA* policy framework on research data). We will aim to process all requests within two weeks.

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

## Acknowledgements

This study was supported by, and received funding from, *UKHSA ACE C215.2 Postpandemic Sensors: Harnessing Isothermal Amplification and Quantum Nanodiamonds for Wastewater Epidemiology* (to DH, DJ, and RAM). This study is also funded by *EPSRC IRC i-sense in Early Warning Sensing Systems for Infectious Diseases and AMR (EP/R00529X/1)* (to DH, ATD, DC, BM, DP, SH, and RAM), the i-sense Next steps award (EP/R018707/1) (to DH, DC, BM, and RAM) and the *EPSRC Digital Health Hub for AMR (EP/X031276/1)* (to SH, and RAM). We thank the LCN for funding studentships ATD, DC and SH. We thank Simon Edwards who managed *ACE* projects and Andrew Singer who chaired the *ACE* programme. We thank UCL researcher Liudmyla Storozhuk for carrying out the carbon black nanoparticle characterisation. This research was also supported by Welsh Government under the Welsh Wastewater Programme (C035/2021/2022) (to JK, KF, RW, and DJ). We thank Tony Harrington and Ian Trick at Dŵr Cymru Welsh Water, UK for their assistance in wastewater sample collection and Bangor laboratory team for their assistance with sample preparation. We thank the Welsh Wastewater Monitoring Group, Bangor and Cardiff, UK for their assistance in the development of sample processing. We thank Gareth Cross and Steve Cobley at Welsh Government for project supervision and management.

## Author contributions

DH, RAM and DJ designed the research and secured funding; DH, ATD, DC, SH performed the research; DP assisted the experiments; DH, ATD, SH analysed the data; DH and RAM drafted the manuscript; JK, KF, RW, DJ provided wastewater samples and support the near-source test, BM, JK, KF, RW contributed to the interpretation of data, all coauthors provided critical revisions to the draft manuscript and approved the final draft.

## Competing interests

B. M. and R.A.M. are inventors on the UK patent application number 1814532.6 filed by University College London Business. The remaining authors declare no competing interests.
