## [Transparent Peer Review file · Nature Communications]

Towards Ultra-sensitive and Rapid Near-source Wastewater-based Epidemiology

Corresponding Author: Professor Rachel McKendry

Version 0:

Reviewer comments:

Reviewer #1

(Remarks to the Author)

This manuscript presents two nanomaterial-based approaches for detecting SARS-CoV-2 RNA in wastewater, aiming to enable near-source detection and advance wastewater-based epidemiology (WBE). While these approaches are worth exploring, several major issues need to be addressed:

1. Turnaround Time: The authors claim their approaches significantly reduce detection time compared to qPCR. However, the time savings primarily come from eliminating sample transportation and using simplified extraction and magnetic bead concentration methods. In theory, if a wastewater facility possesses qPCR equipment or mobile PCR thermocyclers, and adopts similar extraction methods, the turnaround time should be comparable, as qPCR itself is not the time-consuming step. Additionally, the proposed method requires an onsite portable fluorescence reader or microscope, offering no substantial operational advantage. The testing cost comparison should also be considered.
2. Inter-Laboratory Consistency: The rationale for centralized testing (i.e. sampling transportation to a certain lab) is to follow a standardized protocol (SOP) and minimize inter-lab variations. The proposed near-source approach does not address this issue.
3. Detection accuracy in wastewater depends on the recovery efficiency of RNA extraction. The recovery rate and repeatability of the proposed simple extraction and magnetic bead concentration methods should be reported and compared with existing methods.
4. WBE relies on a direct relationship between pathogen gene concentrations (or log-values) and infection levels. However, the proposed method fails to achieve linear or near-linear correlations in Figures 2c, 2e, 3d, 3f, 4b, and 4c, particularly at lower and higher concentrations. This limits its accuracy and applicability.
5. A summary of SARS-CoV-2 RNA concentration ranges in wastewater is needed to justify the analyte concentrations tested. In many cases, concentrations are as low as 2–10 copies/ μ L-reaction (equal to 2–10 copies/mL-wastewater, CT >35), much lower than the tested range. Only 1–2 data points fall within this common range, making it difficult to assess whether the method can detect meaningful concentration changes.
6. The introduction is overly lengthy and does not effectively summarize prior work. The novelty and key challenges addressed by this study should be more clearly articulated.
7. Lines 58–59: What do the authors mean by "WBE is capable of detecting outbreaks in hard-to-reach communities"?

Reviewer #2

(Remarks to the Author)

This manuscript presents a novel approach to wastewater-based epidemiology (WBE) using microwave-modulated nanodiamond fluorescence for detecting viral pathogens. The study explores the potential of spin-based dipstick assays to provide a rapid and sensitive alternative to conventional RT-qPCR methods, addressing the need for near-source testing in wastewater surveillance. The proposed approach offers significant advancements in speed and sensitivity, potentially contributing to early outbreak detection.

However, despite these promising contributions, several critical aspects require substantial revision for clarity, methodological transparency, and proper attribution of prior work. The following review comments outline specific areas that must be addressed before the manuscript can be considered for publication in *Nature Communications*.

(1) The authors reference Miller et al., Nature 2020 as the source of microwave-modulated nanodiamond detection, but this technique was first introduced in Igarashi et al., Nano Letters 2012. Notably, Miller et al. (2020) explicitly cites Igarashi et al.

(2012) as the original work on this method. However, in this paper, Igarashi et al. (2012) is not cited, which may lead readers to mistakenly believe that Miller et al. (2020) is the original source of this technique.

To ensure academic transparency and proper attribution, I strongly recommend that the authors explicitly cite Igarashi et al. (2012) as the foundational work, in addition to Miller et al. (2020). This will ensure that the historical development of the technique is accurately represented and prevent potential misunderstandings.

Furthermore, properly acknowledging prior work is critical for maintaining reproducibility and scientific integrity. By citing Igarashi et al., the authors will help ensure that future researchers can accurately track the evolution of this technique and build upon it correctly.

(2) The manuscript states that the method used to determine the limit of detection (LoD) is based on Miller et al., Nature 2020. However, the manuscript also refers to the use of a four-parameter logistic regression (4PL) model, a term that was not used in Miller et al. (2020), where a linear regression approach combined with the Langmuir adsorption model was instead employed to fit the data.

This raises the following concerns:

1. Is the LoD calculation method exactly the same as in Miller et al. (2020), or has it been modified?

·If the method is exactly the same, the wording should be revised to ensure clarity and avoid confusion.

·If there are differences, these should be explicitly stated, ideally with equations or a direct comparison to the method used in Miller et al. (2020).

2. If modifications were made, how do they impact the LoD calculation?

·If the 4PL model introduces a change, what specific advantage does it provide over the approach used in Miller et al. (2020)?

·If this change affects sensitivity estimation, this should be clearly discussed.

To ensure clarity and transparency, I strongly recommend that the authors either:

•Explicitly confirm that the LoD calculation is identical to Miller et al. (2020) and adjust the wording accordingly, or

•Clearly state how their approach differs from Miller et al. (2020), providing equations and justifications for the modifications.

Clarifying this point will prevent misunderstandings and ensure that the methodology is properly documented for future studies.

(3) In Fig. 3c, the fluorescence intensity time-series plots are shown alongside RPA-dipstick-FND test line images. However, the manuscript does not explicitly state whether these plots are based on real experimental data or if they are schematic representations.

To ensure transparency and reproducibility, I recommend that the authors clarify:

1. Are these plots derived directly from measured data, or are they conceptual illustrations?

·If they are based on real data, specifying which dataset was used and how the data was processed would be helpful.

·If they are schematic, this should be explicitly mentioned in the figure legend to avoid confusion.

2. If the plots are schematic, should actual representative time-series data be provided?

If the plots in Fig. 3c are merely conceptual, I strongly recommend that the authors include typical fluorescence intensity time-series data elsewhere (e.g., in the Supplementary Information). This would improve transparency and allow readers to assess the consistency and reproducibility of the results.

Clarifying this point will improve the credibility of the study and help ensure that future researchers can accurately interpret and replicate the findings.

(4) The manuscript suggests that conventional lateral flow assays (LFAs) often lack sufficient sensitivity and that PCR is considered highly sensitive and the gold standard for WBE. However, it does not specify the quantitative sensitivity threshold required for WBE or how the proposed method compares to existing standards.

To improve clarity and strengthen the manuscript, I strongly recommend:

•Defining a clear sensitivity threshold for WBE.

•Providing a direct numerical comparison between the LoD of the proposed method and qPCR-based WBE approaches.

•Discussing how this method aligns with or deviates from established sensitivity benchmarks.

1. Lack of defined sensitivity criteria for WBE

·While the manuscript discusses the limitations of LFAs and the high sensitivity of PCR, it does not provide specific numerical thresholds for WBE sensitivity.

·Many WBE studies report qPCR LoD values in the range of 1 copy/mL to 10 copies/L, depending on sample processing methods.

·The authors should explicitly define the minimum LoD required for effective WBE and compare it to established guidelines (e.g., WHO/CDC) or previous studies.

2. Insufficient evaluation of whether the proposed LoD meets WBE sensitivity requirements

·The study reports an LoD of 7 copies/reaction (FND assay) and 71 copies/reaction (CBNP assay), but it is unclear whether these values are sufficient for WBE applications.

·A direct comparison with qPCR and other molecular diagnostic methods used in WBE would help clarify the method's suitability.

3. Comparisons with existing studies and guidelines would strengthen the manuscript

·Comparing the LoD of the proposed method with values reported in prior WBE studies or official guidelines would strengthen the manuscript.

·Specifically, comparing the study's LoD (7 copies/reaction) to the minimum detection sensitivity required for qPCR-based WBE (e.g., >1 copy/mL) would enhance the credibility of the study's claims.

Clarifying these points would enhance the manuscript's credibility and provide a stronger justification for the proposed method.

(5) The manuscript presents a promising approach for wastewater epidemiology monitoring using microwave-modulated nanodiamond fluorescence. However, several key experimental parameters required for reproducibility are missing.

1. Excitation beam parameters:

The manuscript does not specify the excitation beam diameter and power (or power density) at the sample surface. These parameters are crucial for understanding the fluorescence intensity and ensuring reproducibility.

2. Microwave power and irradiation method:

While the manuscript discusses microwave modulation, it lacks details on the applied microwave power (dBm or W) and the positioning of the microwave resonator relative to the sample. These factors significantly impact ODMR contrast and overall sensitivity.

3. Photon count and ODMR contrast:

The manuscript does not provide absolute fluorescence photon counts or ODMR contrast values, which are necessary to compare the sensitivity of this method with other detection techniques.

4. Nanodiamond specifications:

The manuscript states that 600 nm FNDs were used, but it does not specify the manufacturer, product number, or NV center concentration.

·If the FNDs were purchased, the supplier and product specifications should be provided for reproducibility.

·If the NV centers were introduced in-house, details on the formation process, NV concentration, and characterization (e.g., ODMR linewidth, spin coherence time) should be included.

·The supplementary information lacks details on the verification of surface chemical and physical properties (e.g., FTIR, XPS, and/or zeta potential) before and after antibody conjugation.

To enhance reproducibility and facilitate comparison with existing methods, I strongly recommend including these experimental details either in the main text or as supplementary information. This will ensure that the reported sensitivity is well-documented and can be effectively benchmarked against other techniques.

(6) This study presents three main reasons for the necessity of near-source measurement:

1. Rapid outbreak detection,
2. Addressing variability in wastewater samples,
3. Logistical and cost improvements.

While points 1. and 3. are well understood, further discussion is needed for point 2.

Specifically, the manuscript mentions challenges such as RNA degradation and fluctuations in viral concentration in wastewater. However, if RT-RPA is performed near the source, the amplified DNA is more stable than RNA. This raises the question of whether measurements could still be conducted in a laboratory after sample transportation without significant issues.

If, as the study suggests, near-source measurement remains essential even after DNA amplification, the authors should provide a clearer justification. To strengthen this argument, I recommend adding further explanations on the following points:

1. Stability of Amplified DNA

·Does the amplified DNA remain stable even after environmental exposure (e.g., temperature variations, chemical/biochemical influences)?

·If DNA stability is a concern, providing experimental data or citing relevant literature on DNase activity in wastewater would help support this claim.

2. Potential Sensitivity Reduction in Laboratory Measurements

·Does a delay in measurement after amplification result in a lower limit of detection (LoD)?

·For example, does prolonged storage of amplified DNA affect fluorescence signal intensity?

·If near-source measurement is crucial for maintaining sensitivity, supporting evidence (e.g., degradation studies, storage experiments) should be provided.

To strengthen the study's argument for near-source measurement, I recommend adding discussions on the stability of amplified DNA and the risk of reduced sensitivity in laboratory measurements. This will provide a clearer justification for the necessity of conducting measurements near the source and improve the overall clarity of the manuscript.

(7) The manuscript states that the RPA-dipstick-CBNP assay requires 40 minutes of amplification, whereas the RPA-dipstick-FND assay requires only 25 minutes. However, it is not explicitly discussed why such a large difference in amplification time exists between the two approaches. The authors should clarify whether this difference is due to fundamental differences in signal generation, nanoparticle interaction, or fluorescence readout sensitivity. If the amplification time was optimized differently for each assay, a brief justification should be provided.

In addition, this study employs two different RNA extraction methods: PEG precipitation and magnetic bead extraction. However, the rationale for selecting each method is not clearly explained. Additionally, PEG precipitation is used for the CBNP assay, while magnetic bead extraction is applied to the FND assay, but there is insufficient discussion on how this difference affects the limit of detection (LoD) and sensitivity. I strongly recommend that the authors clearly justify the choice of extraction methods for each assay and discuss whether these differences influence the study's conclusions. Furthermore, experimental data or references should be provided to demonstrate that the choice of RNA extraction method does not bias the results.

The proposed techniques offer a rapid and sensitive alternative to conventional RT-qPCR, which could enhance near-source outbreak detection. However, as outlined in the review comments, several major revisions are necessary to strengthen the manuscript's scientific rigor and clarity. Given these required revisions, I recommend that the authors address these concerns thoroughly and submit a revised manuscript for reconsideration. With these improvements, the study could make a significant contribution to the field of WBE and may become suitable for publication in *Nature Communications*.

(Remarks to the Author)

The authors have developed and tested a number of rapid dipstick style assays for the detection of SARS-CoV-2 in wastewater, comparing the sensitivity and specificity of detection with RT-qPCR. This is a strong scientific advancement, given the current limitations around transporting wastewater samples to laboratories before analysis and reporting. The experiments and analyses are well designed and well reported. I have a few suggestions and questions, primarily around the framing:

1) Abstract, Line 20 – “RT-qPCR” might be too prescriptive. I would just say PCR as many programs have moved to digital droplet PCR.

2) Line 50 – switch monkeypox to mpox.

3) Lines 73-74 – digital droplet PCR is the gold standard in the US. Cultured virus is the gold standard for polio environmental surveillance.

4) The authors compare sensitivity and specificity of their dipsticks to RT-qPCR and on the model pathogen SARS-CoV-2. Digital droplet PCR is much more sensitive to detect pathogens in wastewater, and at least the US-based programs are all switching to DD-PCR. It would be important to mention this limitation in the discussion. Additionally, SARS-CoV-2 is very abundant in wastewater – much more so than polio, influenza, and RSV. The only other pathogen that gets similar detection rates is norovirus. This will have implications for generalizing the technology to other pathogens. For example, in NY we estimated that polio surveillance in wastewater was much less sensitive than SARS-CoV-2 surveillance. For example, see: Larsen DA, Hill D, Zhu Y, Alazawi M, Chatila D, Dunham C, Faruolo C, Ferro B, Godinez A, Hanson B, Insaf T. Non-detection of emerging and re-emerging pathogens in wastewater surveillance to confirm absence of transmission risk: A case study of polio in New York. *PLOS global public health*. 2024 Dec 31;4(12):e0002381.

Version 1:

Reviewer comments:

Reviewer #1

(Remarks to the Author)

The authors have addressed my comments.

Reviewer #2

(Remarks to the Author)

I would like to commend the authors for their comprehensive and thoughtful revisions in response to the previous round of review. The revised manuscript successfully addresses all major concerns I previously raised, including proper attribution of prior work, clarification of the LoD calculation methodology, inclusion of essential experimental parameters to ensure reproducibility, and a strengthened justification for near-source measurements. These improvements collectively enhance the scientific rigor, transparency, and overall clarity of the study.

Particularly noteworthy are the explicit citations added (e.g., Igarashi et al., 2012), the clarification of methodological distinctions from Miller et al. (2020), and the detailed reporting of key experimental parameters such as excitation beam characteristics, microwave power, and nanodiamond specifications—demonstrating the authors' strong commitment to reproducibility. Moreover, the authors now clearly benchmark the sensitivity of their assay against qPCR-based methods, reinforcing the relevance of their approach in the context of wastewater-based epidemiology. The clarification regarding the use of four-parameter logistic regression (4PL), substantiated by Miller et al. (2022), further strengthens the statistical robustness of the manuscript.

Overall, the revised version represents a substantial improvement and is scientifically sound. I have no remaining concerns and strongly support its publication in *Nature Communications*.

Reviewer #3

(Remarks to the Author)

The authors have done well to respond to the questions and suggestions I had about the manuscript. This is a strong scientific advancement in the field of wastewater and environmental surveillance.

Huang et al. "Towards Ultra-sensitive and Rapid Near-source Wastewater-based Epidemiology"

Response to Reviewer's Comments.

We thank all three Reviewers for their time and thoughtful evaluation on our revised manuscript.

Reviewer 3 highlights our '**strong scientific advancement**' and that our '**experiments and analyses are well designed and well reported**'. **Reviewer 2** states our 'manuscript presents a **novel approach to wastewater-based epidemiology (WBE)** ... [and] **significant advancements in speed and sensitivity**, potentially contributing to early outbreak detection'. They go onto recommend revisions, which we have now thoroughly completed, and 'With these improvements, the study could make a significant contribution to the field of WBE and may become suitable for publication in *Nature Communications*'.

Reviewer 1 raises a number of suggestions which we have rigorously addressed and added new tables, analysis and figures. This constructive feedback has helped us to significantly improve the quality and clarity of our paper.

We have carefully considered all comments and provide a detailed point-by-point response to the Reviewers' comments and amended our manuscript. We believe the manuscript has been significantly improved and hope that it will be accepted for publication in *Nature Communications*.

Reviewer #1:

Reviewer #1 Overall comments: This manuscript presents two nanomaterial-based approaches for detecting SARS-CoV-2 RNA in wastewater, aiming to enable near-source detection and advance wastewater-based epidemiology (WBE). While these approaches are worth exploring, several major issues need to be addressed:

Reviewer #1 Comment 1: Turnaround Time: The authors claim their approaches significantly reduce detection time compared to qPCR. However, the time savings primarily come from eliminating sample transportation and using simplified extraction and magnetic bead concentration methods. In theory, if a wastewater facility possesses qPCR equipment or mobile PCR thermocyclers, and adopts similar extraction methods, the turnaround time should be comparable, as qPCR itself is not the time-consuming step. Additionally, the proposed method requires an onsite portable fluorescence reader or microscope, offering no substantial operational advantage. The testing cost comparison should also be considered.,

Response:

We thank the Reviewer for highlighting the importance of testing times. Our team have an outstanding track record leading national wastewater surveillance for Wales during the pandemic and therefore have a deep understanding of the needs for near-source wastewater-based epidemiology (WBE) testing. The main advantage of our approach and workflow is that the sample process can be performed not only at wastewater treatment plants but also in near-source settings such as prisons, hospitals, and other health care facilities (or within the sewer network at manholes/pumping stations etc) and also in resource limited settings in low-and-middle-income-countries, where PCR costs might be prohibitive. In these settings, early (presymptomatic) detection of cases using WBE approaches would be crucial in reducing the number of cases and preventing nosocomial outbreaks, which often necessitate ward closures and delays in treatment, potentially resulting in loss of life and additional economic burdens. During the pandemic lateral flow tests have been used by millions of people worldwide, in hospitals, workplaces, schools, mass gatherings, borders and in the home, reaching further than any other testing format. The workflow proposed for our nanodiamond enhanced dipstick test is equally portable for near source WBE testing enabling mobile and proactive monitoring - something that would not be feasible with a real-time PCR

machine, which has a much larger footprint and is sensitive to vibrations and agitation. We acknowledge that some approaches for on-site testing at wastewater treatment plants do exist (e.g. Cepheid® LuminUltra®), however, these remain much more expensive than our approach (typically costing £10,000-£20,000 for the equipment alone, see table below – mobile PCR). Our approach is designed to be completely field-portable, low cost and requires minimal training, allowing operation by personnel without specialised technical expertise. This makes it deployable in a wider range of use-cases with advantages including higher sensitivity, faster detection times, greater environmental stability, and reduced contamination risk - making it robust for applications in challenging environments.

Our approach has additional advantages over conventional qPCR-based methods and requires significantly lower temperatures (37-42°C), eliminating the need for thermal cycling equipment and reducing both instrumentation complexity and energy requirements. We have also demonstrated the potential with equipment-less hand warmer bag (~£0.4) for further portability (Cherkaoui *et al.*, *Biosensors and Bioelectronics* 2021)¹. Moreover, we point out that the level of workforce training needed for PCR is generally considerably higher than for a lateral flow or dipstick test.

In response to the query about readers, herein we present two methods: one based on carbon black that does not require a reader, and one based on fluorescent nanodiamonds requiring a fluorescent reader. We have now included a test cost comparison in the SI (Table.S2, shown below), referring to data in our previous published papers Cherkaoui *et al.*, *Biosensors and Bioelectronics* 2021¹, and Miller *et al.*, *Nature* 2020².

Amendments to our manuscript:

A new table is added to Supplementary Information:

Supplementary Information Lines 21-29:

	Dipstick			PCR		
	Carbon Nano Particles	FND – Axxin bespoke portable reader	FND – Smartphone reader (Miller et al. , Nature 2020)	qPCR	ddPCR	Mobile PCR
Reagents cost (£, per test)	8.2	7.4	7.4	20-120	100-140	20-120
Instrument cost (£, reusable)	0	£6,700	£400 + Smartphone	£25,000 - £50,000	£65,000-£150,000	£10,000 - £20,000

Note: all costs were estimated in Sterling Pounds (£). PCR cost estimates are based on a combination of publicly available information and direct quotations received from UK suppliers. "

"SI Table S2: A summary of the estimated costs of our prototype assay, and the comparison to the conventional assays. The listed costs for our dipstick assays are for prototyping and small orders. The Axxin reader was a bespoke version from the company with higher price than the market version. Many of these costs could be dramatically reduced by the economies of scale of mass-manufacturing."

Abstract Text Lines 21-22: "... prompting growing interest in rapid, near-source tests for resource-limited settings."

Abstract Text Lines 31-32: "A proof-of-concept lab-in-a-suitcase nanodiamond assay tested raw, unprocessed wastewater samples."

Main Text Lines 393-408: "The main advantage of our approach and workflow is that the sample process can be performed not only at wastewater treatment plants but also in near-source settings such as prisons, hospitals, and other health care facilities (or within the sewer network at manholes/pumping stations etc) and also in resource limited settings in low and middle income countries, where PCR costs might be prohibitive. In these settings, early (presymptomatic) detection of cases using WBE approaches would be crucial in reducing the number of cases and preventing nosocomial outbreaks, which often necessitate ward closures and delays in treatment, potentially resulting in loss of life and additional economic burdens. The workflow proposed for our nanodiamond enhanced dipstick would readily fit inside a small van, enabling mobile and proactive monitoring - something that would not be feasible with a real-time PCR machine, which is sensitive to vibrations and agitation. We acknowledge that some approaches for on-site testing at wastewater treatment plants do exist (e.g. Cepheid® LuminUltra®), however, these remain significantly more expensive (£10,000-20,000, SI table S2) than our reader and our approach is designed to be much more field-portable, low cost and deployable in a wider range of use-cases (see below). Moreover, we point out that the level of training needed for PCR is generally considerably higher than for a lateral flow or dipstick test."

Reviewer #1 Comment 2: Inter-Laboratory Consistency: The rationale for centralized testing (i.e. sampling transportation to a certain lab) is to follow a standardized protocol (SOP) and minimize inter-lab variations. The proposed near-source approach does not address this issue.

Response:

We thank the Reviewer and agree that Standardised Operating Protocols (SOPs) are important for consistency of any test, however, we wish to highlight that this is an emerging field and to date these have not yet been developed for any WBE assay laboratory or near source (although an ISO standard is being developed for central lab testing). Currently, many different approaches exist for wastewater sample concentration (e.g. ultrafiltration, precipitation, filtration, ultracentrifugation, magnetic bead separation), RNA extraction and quantification (qPCR and ddPCR with several different targets) for COVID-19 WBE globally. The key strategy for quality assurance has been the use of positive and negative controls and sample replicates to assess cross-contamination and sensitivity/accuracy of the produced data. The use of such control measures can be easily implemented in our testing method with minimal effect on the time required for analyses and the associated costs. We have addressed this issue in the revised manuscript.

Amendments to our manuscript:

Main Text Lines 431-438: "In future, we aim to field-test the 'lab-in-a-suitcase' concept with diverse end-users, developing standardised operating procedures (SOP), that include quality control protocols such as preserving test strips for laboratory verification or performing replicate testing to ensure reproducibility. In future such SOPs could incorporate standardised freeze-dried reagent kits with built-in controls to minimise variability across testing sites, while also exploring the integration of readers, modulation and smartphones with data connectivity for real-time geospatial data collection, automated analysis and potential remote quality control oversight."

Reviewer #1 Comment 3: Detection accuracy in wastewater depends on the recovery efficiency of RNA extraction. The recovery rate and repeatability of the proposed simple extraction and magnetic bead concentration methods should be reported and compared with existing methods.

Response:

We thank the Reviewer for this important comment regarding recovery efficiency. We wish to highlight that a comparison between magnetic bead extraction and PEG precipitation methods is already included in our Supplementary Information (Fig.S7c) (see below). This analysis compared both extraction methods across 30 raw wastewater samples, with subsequent qPCR analysis using identical primers. The resulting Ct value distribution plots demonstrate comparable (no significant difference, $P=0.2254$, $t=1.239$, paired two-tailed t -test) recovery efficiencies between our simplified magnetic bead method and conventional PEG precipitation. The magnetic bead method offers significant advantages for near-source applications in terms of speed (40 minutes vs. overnight for PEG), simplicity (fewer steps), and field applicability. To improve clarity, we have enhanced the Fig.S7c caption and incorporated explanatory text in the Methods section to better highlight this comparative analysis of extraction efficiencies between the two methods.

Amendments to our manuscript:

Methods Lines 601-607: "To assess extraction efficiency, we conducted a comparative analysis between our magnetic bead method and conventional PEG precipitation across 30 raw wastewater samples. Both methods were evaluated using identical qPCR primers and conditions. The magnetic bead extraction demonstrated comparable recovery efficiency (two-tailed t -test $P=0.2254$, $t=1.239$), shown in Fig.S7c. This comparison confirms that our streamlined magnetic bead extraction provides suitable recovery rates for reliable near-source detection while significantly reducing processing complexity and time."

Supplementary Information Fig.S7c Line 84: "

Supplementary Information (Fig.S7c captions) Lines 90-97: "..... c) Comparative analysis of RNA extraction efficiency between magnetic bead extraction and PEG precipitation methods. The graph shows qPCR C_t values for 30 paired wastewater samples processed using identical primers and methods (left). Statistical analysis using a paired two-tailed t -test revealed no significant difference in recovery efficiency between the two methods ($P=0.2254$, $t=1.239$) (right). While the magnetic bead method shows more variable recovery rates, it offers significant advantages in processing time, workflow simplicity, and field applicability, which is suitable for near-source testing applications."

Reviewer #1 Comment 4: WBE relies on a direct relationship between pathogen gene concentrations (or log-values) and infection levels. However, the proposed method fails to achieve linear or near-linear correlations in Figures 2c, 2e, 3d, 3f, 4b, and 4c, particularly at lower and higher concentrations. This limits its accuracy and applicability.

Response:

We thank the Reviewer for this observation. We wish to highlight, however that our assay has the necessary sensitivity for early detection of cases, particularly in a near-source setting. Our assay is semi-quantitative which may suffice for many applications (e.g., if a binary positive/negative result is needed or is often sufficient to trigger further investigation), with stronger linearity in the middle-low concentration ranges. In future we could move to quantitative assays for example measuring binding kinetics (the work in our group by Miller *et al.*, have initial the attempts on this direction)³. The sensitivity of the assay can be further enhanced by refining the sample concentration method (e.g. increase sample volume, use portable ultrafiltration device to achieve better viral recoveries). There is inherent variability in wastewater which can contribute to non-linearity, caused by individual shedding rates, dilution effects, RNA degradation, and assay detection limits, among other things. The manuscript was revised to acknowledge this comment and address the discussion on future development on a quantitative assay.

Amendments to our manuscript:

Main Text Lines 348-360: "While our assay demonstrates a concentration-dependent response, it is important to note that it is primarily semi-quantitative in nature (same applied to CNP assay and FND-microscopy assay). The stronger linearity observed in the middle concentration ranges is sufficient for many WBE applications, particularly for early outbreak detection where binary positive/negative results can trigger appropriate investigation protocols. The inherent variability in wastewater matrices - including fluctuating individual shedding rates, dilution effects, and RNA degradation - can contribute to the non-linear response observed at very low and very high viral concentrations. Our experience with national WBE programmes suggests that this level of quantification is adequate for near-source applications focused on early warning capabilities. Future refinements could enhance quantitative performance through improved sample concentration methods (e.g., increased sample volumes or portable ultrafiltration) and advanced binding kinetics analysis, building on our previous work⁵²."

Reviewer #1 Comment 5: A summary of SARS-CoV-2 RNA concentration ranges in wastewater is needed to justify the analyte concentrations tested. In many cases, concentrations are as low as 2–10 copies/μL-reaction (equal to 2–10 copies/mL-wastewater, CT >35), much lower than the tested range. Only 1–2 data points fall within this common range, making it difficult to assess whether the method can detect meaningful concentration changes

Response:

We thank the Reviewer for this important point regarding SARS-CoV-2 concentration ranges in wastewater. To address this concern, we would like to point to research by Wade *et al.* from the UK Health Security Agency who led the WBE programme. Wade *et al.* show that typical SARS-CoV-2 viral fragment concentrations in wastewater during the pandemic ranged primarily from 10² to 10⁵ genome copies per litre (gc/L) (0.1-100 gc/mL) (see below the Fig.2 from the paper)⁴. Detection sensitivity in WBE applications relies on effective sample concentration and RNA extraction protocols that maximise recovery from original wastewater samples. In our study, we have developed and validated rapid concentration and extraction protocols specifically optimised for near-source WBE applications, balancing efficiency with

field practicality. Our method processes 50 mL of original wastewater and concentrates it to 50 μ L of extracted RNA. The FND method demonstrated a detection limit of 7 copies per reaction (95% CI: 3-13 copies) using standardised RNA samples. When calculating the theoretical detection limit in original wastewater - assuming ideal recovery efficiency through our concentration and extraction steps - this translates to approximately 140 gc/L (95% CI: 60-260 gc/L), the low end of the reported range, and well below the Reviewer's low value of 2000 gc/L (2 gc/mL). In practice, variable RNA recovery efficiencies from complex wastewater matrices may affect this limit, though our comparative analysis with conventional methods showed similar overall recovery rates (see Methods-Magnetic Beads wastewater extraction, and Fig.S7c). Even accounting for this variability, our sensitivity falls within the lower range of reported concentrations needed for early outbreak detection. Our successful evaluation using actual raw wastewater samples from a national Welsh surveillance program confirms the method's effectiveness for practical WBE implementation. This sensitivity is comparable to our qPCR work in national WBE programmes (Farkas *et al.*, 2021)⁵, which achieved a Limit of Detection of 0.92 gc/ μ L RNA, equivalent to 611 gc/L wastewater, when processed through conventional concentration steps. We have revised the manuscript to clarify this important relationship between our assay's detection limit and typical viral concentrations in wastewater by adding explanatory text in the Results section and relevant reference.

Fig. 2. An example comparison of SARS-CoV-2 concentrations and SARS-CoV-2 loads per capita measured from wastewater sampled at an English sewage treatment works, showing (a) variation over time; and (b) correlation between the two metrics.⁴

Amendments to our manuscript:

Methods Lines 599-601: "Detection sensitivity in WBE applications relies on effective sample concentration and RNA extraction protocols that maximise recovery from original wastewater samples."

Main Text Lines 325-336: "Our workflow processes 50 mL of raw wastewater down to 50 μ L of extracted RNA, with the FND method's detection limit of 7 copies per reaction (95% CI: 3-13 copies), based on the standardised RNA samples, translating to approximately 140 genome copies per litre, gc/L (95% CI: 60-260 gc/L), in the original wastewater sample, assuming ideal RNA recovery efficiency. This sensitivity falls within the lower range of SARS-CoV-2 RNA concentrations typically reported in WBE studies (10^2 - 10^5 gc/L) during the pandemic⁵⁰, making it suitable for early outbreak detection. Notably, this detection capability is comparable to established laboratory qPCR methods used in national WBE surveillance programmes, which achieve limits of detection around 611 gc/L when processed through conventional concentration steps with similar RNA recovery efficiency⁵¹ (see Methods and Fig.S7c). The ability to achieve this sensitivity level with a field-deployable platform represents a significant advancement for near-source WBE applications."

Reviewer #1 Comment 6: The introduction is overly lengthy and does not effectively summarize prior work. The novelty and key challenges addressed by this study should be more clearly articulated.

Response: We thank the Reviewer and have updated the introduction with additional references and also aimed to further articulate the novelty and key challenges addressed by our near-source assay for resource limited settings.

Amendments to our manuscript:

We have revised the introduction by removing approximately 300 words to make it more concise and focused. We have also rewritten the final paragraph to more clearly articulate the novelty of our approach. The revisions span throughout the introduction section (**Lines 36-126**), with particular emphasis on:

Main Text Lines 112-126: "Herein we report new approaches in near-source WBE through the development of two complementary nanoparticles-based rapid lateral flow detection platforms, focusing on SARS-CoV-2 as an exemplar. Our innovations include: 1) the first implementation of commercially available carbon black nanoparticles (CBNPs) for visual readout in WBE rapid testing, providing an accessible, equipment-free detection option; and 2) the pioneering application of spin-enhanced FNDs in a dipstick format for wastewater surveillance, achieving ultra-sensitivity in field-deployable pathogen detection. Recognising that near-source WBE requires streamlined workflows, we developed a user-friendly one-pot assay integrating lyophilised RPA, significantly reducing procedural complexity. We further innovated simplified sample concentration and magnetic bead extraction methods, enabling rapid on-site sample processing. The integration of these technologies with a portable nanodiamond reader creates a complete lab-in-a-suitcase solution suitable for deployment at wastewater sources (Fig. 1b-i, iii). To validate this approach, we conducted a comprehensive blinded pilot study using 62 raw wastewater samples from the Welsh national WBE programme, demonstrating real-world applicability and performance."

Reviewer #1 Comment 7: Lines 58–59: What do the authors mean by "WBE is capable of detecting outbreaks in hard-to-reach communities"?

Response:

We thank the Reviewer and clarify this refers to WBE capability to monitor and detect disease outbreaks in communities that are often hard to reach via traditional surveillance, which relies on clinic attendance. Hard to reach communities may be affected by migrant or travelling communities, social stigma, inequalities in access to infrastructural or resource limitations, geographical isolation, or low healthcare utilisation, including the lack of clinical surveillance. For example, in 2022 vaccine-like-type-2 poliovirus isolates were detected in the London Beckton Sewage Treatment plant serving a population of 4 million, despite no cases presenting at clinic. The relevant references (ref 16, 17 in main text)^{6,7} were added to clarify this advantage in WBE.

Amendments to our manuscript:

Main Text Lines 52-55: "Moreover, WBE is capable of detecting outbreaks in communities that are often hard to reach due to social stigma, migrant or travelling communities, healthcare inequalities in access to infrastructural or resource limitations, and low healthcare utilisation regions^{16,17}."

Reviewer #2:

Reviewer #2 Overall comments: This manuscript presents a novel approach to wastewater-based epidemiology (WBE) using microwave-modulated nanodiamond fluorescence for detecting viral pathogens. The study explores the potential of spin-based dipstick assays to provide a rapid and sensitive alternative to conventional RT-qPCR methods, addressing the need for near-source testing in wastewater surveillance. The proposed approach offers significant advancements in speed and sensitivity, potentially contributing to early outbreak detection. However, despite these promising contributions, several critical aspects require substantial revision for clarity, methodological transparency, and proper attribution of prior work. The following review comments outline specific areas that must be addressed before the manuscript can be considered for publication in Nature Communications.

The proposed techniques offer a rapid and sensitive alternative to conventional RT-qPCR, which could enhance near-source outbreak detection. However, as outlined in the review comments, several major revisions are necessary to strengthen the manuscript's scientific rigor and clarity. Given these required revisions, I recommend that the authors address these concerns thoroughly and submit a revised manuscript for reconsideration. With these improvements, the study could make a significant contribution to the field of WBE and may become suitable for publication in Nature Communications.

Response:

We thank the Reviewer for their positive feedback - "The proposed approach offers significant advancements in speed and sensitivity potentially contributing to early outbreak detection", and for the time they have dedicated to reviewing our manuscript. We have taken onboard their constructive comments to improve our manuscript. In the section below, we carefully address each of the Reviewer's comments in turn.

Reviewer #2 Comment 1: The authors reference Miller et al., Nature 2020 as the source of microwave-modulated nanodiamond detection, but this technique was first introduced in Igarashi et al., Nano Letters 2012. Notably, Miller et al. (2020) explicitly cites Igarashi et al. (2012) as the original work on this method. However, in this paper, Igarashi et al. (2012) is not cited, which may lead readers to mistakenly believe that Miller et al. (2020) is the original source of this technique. To ensure academic transparency and proper attribution, I strongly recommend that the authors explicitly cite Igarashi et al. (2012) as the foundational work, in addition to Miller et al. (2020) This will ensure that the historical development of the technique is accurately represented and prevent potential misunderstandings. Furthermore, properly acknowledging prior work is critical for maintaining reproducibility and scientific integrity. By citing Igarashi et al., the authors will help ensure that future researchers can accurately track the evolution of this technique and build upon it correctly.

Response:

We sincerely thank the Reviewer, and are happy to cite prior foundational work (Igarashi et al. (2012), ref 44 in main text) on microwave-modulated nanodiamond detection, to clearly acknowledge the historical development of this technique.

Amendments to our manuscript:

Main Text Lines 101-106: "To overcome this, fluorescent nanodiamonds (FNDs) containing nitrogen-vacancy centres show promise in the emerging field of quantum sensors. Their spin-dependent emission intensity can be modulated using a microwave field⁴⁴. They have been shown to improve detection limits by 100,000 with model biotin-avidin assays and 7,500-fold over gold nanoparticles for amplicon detection, improving signal-to-noise ratio (SNR) on lateral flow test by background separation⁴⁵."

Reviewer #2 Comment 2: The manuscript states that the method used to determine the limit of detection (LoD) is based on Miller et al., Nature 2020. However, the manuscript also refers to the use of a four-parameter logistic regression (4PL) model, a term that was not used in Miller et al. (2020), where a linear regression approach combined with the Langmuir adsorption model was instead employed to fit the data.

This raises the following concerns:

1. Is the LoD calculation method exactly the same as in Miller et al. (2020), or has it been modified?

·If the method is exactly the same, the wording should be revised to ensure clarity and avoid confusion.

·If there are differences, these should be explicitly stated, ideally with equations or a direct comparison to the method used in Miller et al. (2020).

2. If modifications were made, how do they impact the LoD calculation?

·If the 4PL model introduces a change, what specific advantage does it provide over the approach used in Miller et al. (2020)?

·If this change affects sensitivity estimation, this should be clearly discussed.

To ensure clarity and transparency, I strongly recommend that the authors either:

•Explicitly confirm that the LoD calculation is identical to Miller et al. (2020) and adjust the wording accordingly, or

•Clearly state how their approach differs from Miller et al. (2020), providing equations and justifications for the modifications.

Clarifying this point will prevent misunderstandings and ensure that the methodology is properly documented for future studies.

Response:

Response to point 1:

We appreciate the Reviewer highlighting this and can confirm the fitting method used was reported in Miller *et al.* (2022)⁸, which builds on a method proposed by Holstein *et al.* (2015)⁹. We used an open-source fitting software tool, developed by Miller, and available on GitHub (<https://github.com/bensmiller/detection-limit-fitting>). We have updated the **Methods Lines 558-576** to correct this error.

Response to point 2:

The updated LoD calculation, published in Miller *et al.* (2022)⁸ differs in three ways from the previously published method published in Miller *et al.* (2020)²:

1. It uses a different variance to calculate the LoD:
 - a. The earlier method (2020) defined the LoD as the intersection of the upper 95% confidence interval of the negative controls with the lower 95% confidence interval of the model fit, thereby accounting for variance in the negative control and variance in the fit.
 - b. In the subsequent work (2022), the LoD was defined as the intersection of upper 95% confidence interval of the negative controls with the lower 95% confidence interval of the lowest detectable concentration (a sample exactly at the LoD). The latter confidence interval was calculated using a characteristic variance for positive samples: the average

pooled variance of the positive samples. To facilitate this, a log transform of the x and y data was used to normalise variances, allowing all y-values to be used to calculate the variance, not just those close to the LoD. Please see image below from Holstein *et al.* (2015)⁹:

2. The Holstien paper further proposes a method for calculating the standard error of the LoD estimate, crucially allowing for statistically robust comparisons between LoDs and the calculation of confidence intervals. It includes variance from signal measurements of the blanks and the positive controls, and the model fitting uncertainty.
3. In this work we've fitted to a stretched exponential model, which shows a slightly better fit than the 4PL (as per the AICc). This is updated from the original manuscript, where the 4PL was used. See below for equation. This model is available with the open source software tool available at [GitHub - bensmiller/detection-limit-fitting: Software for fitting robust detection limits to serial dilution data](https://github.com/bensmiller/detection-limit-fitting).

There are a two additional smaller changes:

- (i) the t distribution was used (for small n) instead of the Gaussian approximation used previously.
- (ii) A G test was used to exclude outlier variances when calculating the average pooled variance.

Although the two methods are very similar (and indeed give very similar LoDs), the rationale for this change was threefold:

- (i) This choice of the positive sample pooled variance rather than the fit variance was a more intuitive definition of LoD.
- (ii) It is a more conservative LoD, giving slightly higher values in general.
- (iii) We were able to calculate 95% confidence intervals of the LoDs, quoted in this work.

Formally, the concentration, C , and lock-in signal, y , data are log transformed identically:

$$y^* = \log_{10}(y + 2)$$

$$C^* = \log_{10}(C + 2)$$

The y-value corresponding to the upper 95% confidence interval of the negative controls, L_c , is defined as:

$$L_c = \bar{y}_c^* + t(0.95, n - 1)s_c$$

where \bar{y}_c^* is the mean of the negative controls, t is the Student's t inverse cumulative distribution function, n is the number of negative control, and s_c is the sample standard deviation of the negative controls y_c^* .

The characteristic variance for the positive samples, s_t , is calculated by averaging the variances of the log-normalised y-data:

$$s_t = \sqrt{\frac{\sum_{i=1}^m s_i^2}{m}}$$

where s_i is the sample standard deviation for repeats at a single tested concentration, and m is the number of concentrations tested.

The y -value corresponding to the LoD, L_D , is defined as:

$$L_D = L_C + t(0.95, m(n-1))s_t$$

The log-transformed data are then fitted to a stretched exponential model:

$$y(C) = a + b \left(1 - e^{-\left(\frac{C}{k}\right)^d} \right)$$

$$y^*(C^*) = \log_{10} \left[a + b \left(1 - e^{-\left(\frac{10C^*-2}{k}\right)^d} \right) \right]$$

where a , b , k , and d are fitted parameters.

The LoD is defined as the value of C at the intersection of the mean of the fit with L_D : $y^*(C^*) = L_D$.

The confidence interval of the LoD is defined as in Holstein *et al.* (2015)⁹.

Amendments to our manuscript:

Methods Lines 560-578: "FND Assay Limit of Detection Analysis. A statistical method for determining the LoD was employed based on previously reported methods reported⁵⁵. The LoD is defined as the intersection of upper 95% confidence interval of the negative controls with the lower 95% confidence interval of the lowest detectable concentration (a sample exactly at the LoD). The latter confidence interval was calculated using a characteristic variance for positive samples: the average pooled variance of the positive samples. To facilitate this, a log transform of the concentrations, C , and lock-in signals, y , was used to normalise variances, allowing all y -values to be used to calculate the variance, not just those close to the LoD:

$$y^* = \log_{10}(y + 2)$$

$$C^* = \log_{10}(C + 2)$$

See Miller *et al.* (2022)⁵⁵ and Holstein *et al.* (2015)⁵⁶ for further details. The data were then fitted to a stretched exponential model:

$$y(C) = a + b \left(1 - e^{-\left(\frac{C}{k}\right)^d} \right)$$

$$y^*(C^*) = \log_{10} \left[a + b \left(1 - e^{-\left(\frac{10C^*-2}{k}\right)^d} \right) \right]$$

where a , b , k , and d are fitted parameters. This model was selected as the best fit based on the root mean squared errors (RMSE) and the small-sample corrected Akaike Information Criterion (AICc) parameters, as described in Miller *et al.* (2022)⁵⁵. The fitting software is open source and available at GitHub (<https://github.com/bensmiller/detection-limit-fitting/>). The confidence interval of the LoD is defined as in Holstein *et al.* (2025)⁵⁶."

Reviewer #2 Comment 3: In Fig. 3c, the fluorescence intensity time-series plots are shown alongside RPA-dipstick-FND test line images. However, the manuscript does not explicitly state whether these plots are based on real experimental data or if they are schematic representations.

To ensure transparency and reproducibility, I recommend that the authors clarify:

1. Are these plots derived directly from measured data, or are they conceptual illustrations?

·If they are based on real data, specifying which dataset was used and how the data was processed would be helpful.

·If they are schematic, this should be explicitly mentioned in the figure legend to avoid confusion.

2. If the plots are schematic, should actual representative time-series data be provided?

If the plots in Fig. 3c are merely conceptual, I strongly recommend that the authors include typical fluorescence intensity time-series data elsewhere (e.g., in the Supplementary Information). This would improve transparency and allow readers to assess the consistency and reproducibility of the results.

Clarifying this point will improve the credibility of the study and help ensure that future researchers can accurately interpret and replicate the findings.

Response:

We are happy to clarify the time-series plots shown in Fig. 3c are based on our real experimental data.

Response to point 1:

These plots are actual experimental data from our measurements. It is a comparison between lock-in and conventional fluorescence images analysis. On the left, it is the fluorescence images captured from lateral flow test strips, throughout series dilution of analyte concentration. On the right, it is the intensity-time plots represent signal modulation (pixel variation). The figure caption was revised to clarify the plots.

Response to point 2:

We are pleased to include extended fluorescence intensity time-series plots in the supplementary information (Fig. S4) to demonstrate reproducibility and consistent accessibility of the results.

Amendments to our manuscript:

Main Text Lines 246-247: "The microwave signal modulation approach enables lock-in detection with a lower LoD than imaging the test line (Fig. 3c, Fig. S4)."

Figure 3 Caption Text Lines 271-276: "**c**) Comparison between lock-in and conventional fluorescence images analysis. Captures of measured RPA-dipstick-FND test lines (left) from selected concentrations of a serial dilution, with the corresponding fluorescent intensity time series plots (right), demonstrating that signal modulation enables detection well below the concentration required to directly image the test line (see extended plots in Fig. S4)."

Supplementary Information Lines 61-66: "

Fig.S4 Extended comparison between lock-in and conventional fluorescence images analysis. On the top, it is the fluorescence images captured from lateral flow test strips, throughout series dilution of analyte concentration. On the bottom, it is the intensity–time plots represent signal modulation (pixel variation), showing that a periodic signal is still evident after the test line is no longer visible in the fluorescence images.”

Reviewer #2 Comment 4: The manuscript suggests that conventional lateral flow assays (LFAs) often lack sufficient sensitivity and that PCR is considered highly sensitive and the gold standard for WBE. However, it does not specify the quantitative sensitivity threshold required for WBE or how the proposed method compares to existing standards.

To improve clarity and strengthen the manuscript, I strongly recommend:

- Defining a clear sensitivity threshold for WBE.
- Providing a direct numerical comparison between the LoD of the proposed method and qPCR-based WBE approaches.
- Discussing how this method aligns with or deviates from established sensitivity benchmarks.

1. Lack of defined sensitivity criteria for WBE

·While the manuscript discusses the limitations of LFAs and the high sensitivity of PCR, it does not provide specific numerical thresholds for WBE sensitivity.

·Many WBE studies report qPCR LoD values in the range of 1 copy/mL to 10 copies/L, depending on sample processing methods.

·The authors should explicitly define the minimum LoD required for effective WBE and compare it to established guidelines (e.g., WHO/CDC) or previous studies.

2. Insufficient evaluation of whether the proposed LoD meets WBE sensitivity requirements

·The study reports an LoD of 7 copies/reaction (FND assay) and 71 copies/reaction (CBNP assay), but it is unclear whether these values are sufficient for WBE applications.

·A direct comparison with qPCR and other molecular diagnostic methods used in WBE would help clarify the method's suitability.

3. Comparisons with existing studies and guidelines would strengthen the manuscript

·Comparing the LoD of the proposed method with values reported in prior WBE studies or official guidelines would strengthen the manuscript.

·Specifically, comparing the study's LoD (7 copies/reaction) to the minimum detection sensitivity required for qPCR-based WBE (e.g., >1 copy/mL) would enhance the credibility of the study's claims.

Clarifying these points would enhance the manuscript's credibility and provide a stronger justification for the proposed method.

Response:

We thank the Reviewer for this important comment regarding sensitivity thresholds for WBE. We have benchmarked our method against published concentration ranges and conventional qPCR approaches. Our method demonstrates suitable sensitivity for near-source early warning applications, which we address point-by-point below:

Response to point 1:

Defining sensitivity requirements for effective WBE: As of now, there is no standardised minimum detection limit for SARS-CoV-2 in wastewater established by the World Health Organization (WHO) or other global health agencies. The detection limits vary depending on the analytical methods, protocols, and the specific goals of monitoring. Drawing on our expertise from national WBE programmes, we can address this with specific data. Wade et al. reported that typical SARS-CoV-2 viral fragment concentrations in wastewater during the pandemic ranged primarily from 10^2 to 10^5 genome copies per litre⁴ (Please see the response to the Reviewer #1, Comment 5). For early outbreak detection applications, sensitivity in the lower end of this range is essential, particularly when viral shedding is just beginning in a community.

Response to point 2:

Evaluation of LoD for WBE: Detection sensitivity in WBE applications relies on effective sample concentration and RNA extraction protocols that maximise recovery from original wastewater samples. Our FND method demonstrated a detection limit of 7 copies per reaction using standardised RNA samples (71 copies/reaction in CBNP method). When calculating the theoretical detection limit in original wastewater - assuming ideal recovery efficiency through our concentration and extraction steps - this translates to approximately 140 genome copies per litre, gc/L (1420 gc/L in CBNP method). In practice, variable RNA recovery efficiencies from complex wastewater matrices may affect this limit, though our comparative analysis with conventional methods showed similar overall recovery rates (see Methods-Magnetic Beads wastewater extraction, and Fig.S7c). Our successful evaluation using actual raw wastewater samples from a national surveillance program confirms the method's effectiveness for practical WBE implementation.

Response to point 3:

Comparison with established methods: This sensitivity is comparable to our qPCR work in national WBE programmes (Farkas *et al.*, 2021), which achieved an LoD of 0.92 gc/ μ L RNA, equivalent to 611 gc/L wastewater, when processed through conventional concentration steps. Furthermore, the detection limits can vary depending on the analytical methods and protocols used and the choice of method should depend on the goal of the monitoring. For tracking inclines/declines of an abundant pathogen, a less sensitive method suitable for mass testing may be used (Karthikeyan *et al.*, 2021)¹⁰. For early detection of

pathogens, quantification capability becomes less important, and a highly sensitive concentration-detection system should be used (Farkas *et al.*, 2022)¹¹. Our assay is designed primarily for the latter purpose - providing rapid, near-source detection that can serve as an early warning system. Even accounting for practical recovery variability, our sensitivity falls within the range needed for early outbreak detection. We have also noted that while our current approach achieves suitable sensitivity for SARS-CoV-2 detection, the concentration step can be further refined in the future by increasing sample volume or by introducing simple and portable ultrafiltration devices that are also suitable for on-site testing, potentially enhancing sensitivity for pathogens present at lower concentrations.

We have revised the manuscript to clarify this important relationship between our assay's detection limit and typical viral concentrations in wastewater by adding explanatory text in the Results section, with relevant references supporting our sensitivity benchmarks.

Amendments to our manuscript:

Methods Lines 599-601: "Detection sensitivity in WBE applications relies on effective sample concentration and RNA extraction protocols that maximise recovery from original wastewater samples."

Main Text Lines 325-336: "Our workflow processes 50 mL of raw wastewater down to 50 µL of extracted RNA, with the FND method's detection limit of 7 copies per reaction (95% CI: 3-13 copies), based on the standardised RNA samples, translating to approximately 140 genome copies per litre, gc/L (95% CI: 60-260 gc/L), in the original wastewater sample, assuming ideal RNA recovery efficiency. This sensitivity falls within the lower range of SARS-CoV-2 RNA concentrations typically reported in WBE studies (10^2 - 10^5 gc/L) during the pandemic⁵⁰, making it suitable for early outbreak detection. Notably, this detection capability is comparable to established laboratory qPCR methods used in national WBE surveillance programmes, which achieve limits of detection around 611 gc/L when processed through conventional concentration steps with similar RNA recovery efficiency⁵¹ (see Methods and Fig.S7c). The ability to achieve this sensitivity level with a field-deployable platform represents a significant advancement for near-source WBE applications."

Reviewer #2 Comment 5: The manuscript presents a promising approach for wastewater epidemiology monitoring using microwave-modulated nanodiamond fluorescence. However, several key experimental parameters required for reproducibility are missing.

1. Excitation beam parameters:

The manuscript does not specify the excitation beam diameter and power (or power density) at the sample surface. These parameters are crucial for understanding the fluorescence intensity and ensuring reproducibility.

2. Microwave power and irradiation method:

While the manuscript discusses microwave modulation, it lacks details on the applied microwave power (dBm or W) and the positioning of the microwave resonator relative to the sample. These factors significantly impact ODMR contrast and overall sensitivity.

3. Photon count and ODMR contrast:

The manuscript does not provide absolute fluorescence photon counts or ODMR contrast values, which are necessary to compare the sensitivity of this method with other detection techniques.

4. Nanodiamond specifications:

The manuscript states that 600 nm FNDs were used, but it does not specify the manufacturer, product number, or NV center concentration.

·If the FNDs were purchased, the supplier and product specifications should be provided for reproducibility.

·If the NV centers were introduced in-house, details on the formation process, NV concentration, and characterization (e.g., ODMR linewidth, spin coherence time) should be included.

·The supplementary information lacks details on the verification of surface chemical and physical properties (e.g., FTIR, XPS, and/or zeta potential) before and after antibody conjugation.

To enhance reproducibility and facilitate comparison with existing methods, I strongly recommend including these experimental details either in the main text or as supplementary information. This will ensure that the reported sensitivity is well-documented and can be effectively benchmarked against other techniques.

Response: We thank the Reviewer for highlighting the omission of these experimental parameters. We are pleased to confirm the following, and have now included these details to ensure reproducibility:

1. Excitation beam parameters:

- Beam diameter: 2.1 mm at the sample surface
- Power: 60 mW

2. Microwave power and irradiation method:

- Applied microwave power: 16.5 dBm (0.045 W)
- The microwave resonator is positioned directly under the sample, aligned with the test line.
- Distance from resonator to test line: approximately 1 mm (the thickness of the test strip including backing card)

3. Photon count and ODMR contrast:

- We have added a new supplementary figure (Fig. S8) (also see below) showing ODMR measurements
- Typical photon count rates: 3.03×10^5 - 3.44×10^5 counts/s across the range from weak to strong positive test lines. For weak positives, the majority of photons originate from nitrocellulose background fluorescence. The estimated photon contribution from FNDs can be approximated as the difference between strong and weak positive signals ($\approx 4.04 \times 10^4$ counts/s), which represents approximately 12% of the total photon count in strong positive samples.
- ODMR contrast: 0.20% at 1.43 GHz (ΔE^*) and 1.8% at 2.87 GHz (ΔE).

4. Nanodiamond specifications:

- Manufacturer: Adámas Nanotechnologies
- Product number: custom product, 600nm, 1 mg/mL in DI water
- NV centre concentration: approximately 3 ppm
- Surface functionalisation: The FNDs were functionalised with Polyglycerol, (PG)-coated, and verified by Adámas Nanotechnologies. The FND characterisation was included in Figure 3b-iii, showing DLS measurements of hydrodynamic diameter (600 nm) after antibody functionalisation. Additional characterisation methods followed the same protocols described in Miller *et al.* (2020), including FTIR to verify surface chemistry.

We have updated the Methods section to include these experimental parameters.

Amendments to our manuscript:

Method Lines 499-501: "Polyglycerol (PG)-coated 600nm FNDs (custom purchase from Adámas Nanotechnologies, 1 mg/mL in DI water, ~3ppm NV centres) were conjugated to antibodies using disuccinimidyl carbonate (DSC)."

Method Lines 532-558: "Fluorescence measurements and analysis. Optically detected magnetic resonance (ODMR) measurements were carried out on modified fluorescent nanodiamonds. 600nm FND-PG-BSA-biotin was used to run on a polystreptavidin LF strip at a high concentration (1:5 dilution) to generate a strong positive signal test line. The strip was placed directly on the resonator board fixed to the microscope stage with microwaves being supplied using a (SynthUSB3) source and microwave amplifier (ZRL-3500+, Mini Circuits) through a linear strip line to sweep across frequencies from 1 to 3 GHz. The distance from resonator to test line is approximately 1 mm (the thickness of the test strip including backing card). A MATLAB code was used to plot the frequency intensity across different frequencies. (see ODMR measurements in Fig.S8). The dipstick strips were imaged using a fluorescence microscope (Olympus BX51) with a 550nm green LED excitation light source (CoolLED pE-4000) producing 60 mW at sample, a filter cube with an excitation filter (Semrock-500nm bandpass, 49nm bandwidth), a dichroic mirror (Semrock-596nm edge), and 593nm long-pass emission filter (Semrock). A 20x/0.4 BD objective lens producing a 2.1 mm beam diameter at focus was used. Images were recorded using a high-speed camera (Hamamatsu, ORCA-Flash4.0 V3) and HCLImage Live software (Hamamatsu) for 15s at a sample rate of 33.33 frames/sec, where the mean of each frame were calculated to give a time-series of mean pixel values. Typical photon count rates are 3.03×10^5 - 3.44×10^5 counts/s for weak to strong positive testing conditions (approximately 12% of the total photon count $\approx 4.04 \times 10^4$ counts/s contributed from FNDs). A voltage-controlled oscillator (VCO) (Mini-Circuits-ZX95-3360+) and a low-power amplifier (Mini-Circuits-ZX60-33LN+) were connected to an omega-shaped resonator and circuit board (Minitron, Rogers 4003c 0.8 mm substrate with 300 gm^{-2} copper weight) to generate the microwave field (+17 dBm input power at 2.87 GHz). The modulated signal is achieved by modulating the VCO input with a reference frequency generator at 4 Hz, using a 32.768 Hz crystal oscillator (Farnell, DS32KHZ) and 14-stage frequency divider (Farnell, CD4060BM). The microwave circuit and full microscope set up described in Miller *et al.* 2020⁴⁵"

Supplementary Information Lines 99-107: "

Fig.S8 Continuous wave optically detected magnetic resonance (CW-ODMR) measurement of the nitrogen-vacancy (NV) centres in nanodiamond. (Top) The fluorescence signal decreases at approximately 2.87 GHz and 1.43 GHz, corresponding to the ground-state ($\Delta E = 2.87\text{GHz}$) and optically excited-state ($\Delta E^* = 1.43\text{GHz}$) zero-field splittings, respectively. The fluorescence reduction is approximately 0.20% at 1.43 GHz and 1.8% at 2.87 GHz. (Bottom) Zoomed-in spectrum around 2.87 GHz, revealing the detailed structure of the ODMR dip.”

Reviewer #2 Comment 6: This study presents three main reasons for the necessity of near-source measurement:

1. Rapid outbreak detection,
2. Addressing variability in wastewater samples,
3. Logistical and cost improvements.

While points 1. and 3. are well understood, further discussion is needed for point 2.

Specifically, the manuscript mentions challenges such as RNA degradation and fluctuations in viral concentration in wastewater. However, if RT-RPA is performed near the source, the amplified DNA is more stable than RNA. This raises the question of whether measurements could still be conducted in a laboratory after sample transportation without significant issues.

If, as the study suggests, near-source measurement remains essential even after DNA amplification, the authors should provide a clearer justification. To strengthen this argument, I recommend adding further explanations on the following points:

1. Stability of Amplified DNA

·Does the amplified DNA remain stable even after environmental exposure (e.g., temperature variations, chemical/biochemical influences)?

·If DNA stability is a concern, providing experimental data or citing relevant literature on DNase activity in wastewater would help support this claim.

2. Potential Sensitivity Reduction in Laboratory Measurements

·Does a delay in measurement after amplification result in a lower limit of detection (LoD)?

·For example, does prolonged storage of amplified DNA affect fluorescence signal intensity?

·If near-source measurement is crucial for maintaining sensitivity, supporting evidence (e.g., degradation studies, storage experiments) should be provided.

To strengthen the study's argument for near-source measurement, I recommend adding discussions on the stability of amplified DNA and the risk of reduced sensitivity in laboratory measurements. This will provide a clearer justification for the necessity of conducting measurements near the source and improve the overall clarity of the manuscript.

Response:

We thank the Reviewer for this thoughtful comment regarding the necessity of near-source measurements. While amplified DNA remains stable under proper storage conditions, our near-source approach offers several important advantages: significantly faster turnaround time (2 hours vs. 72 hours), reduced costs, improved sample integrity, and increased accessibility for locations without laboratory infrastructure. These practical benefits complement considerations about DNA stability and together provide strong justification for on-site testing. We address the specific points below:

Response to point 1:

Stability of Amplified DNA: The amplified DNA from our RPA reaction remains stable under typical storage conditions. According to the RPA manufacturer (TwistDx), amplicons can be stored at 4°C for short periods and at -20°C for longer periods. DNase activity in wastewater is not a concern for our amplified products since the wastewater sample undergoes lysis and magnetic bead purification before amplification. Additionally, our one-pot reaction within the lab-in-a-suitcase setup minimises the exposure of amplified DNA to environmental conditions that could lead to degradation.

Response to point 2:

Rationale for Near-Source Measurement: While delayed measurement after amplification would not significantly reduce sensitivity, our focus on near-source testing is driven by several key advantages:

- i) **Early outbreak detection and workflow continuity:** Having processed samples and performed amplification on-site, completing the workflow with immediate result readout provides logical continuity and maximum time savings. This early warning is particularly important in an outbreak setting. The test strips could be preserved and transported back to laboratories for quality control verification, as we demonstrated with selected samples. As we develop this technology further in the future, we plan to incorporate standardised protocols for both on-site testing and laboratory verification within a comprehensive standard operating procedure (SOP) to ensure consistency and reliability.
- ii) **Cost efficiency:** By overcoming the need for sample transportation, cold chain maintenance, and centralised laboratory processing, we substantially reduce the overall cost per test.
- iii) **Sample integrity and accessibility:** Processing samples immediately minimises RNA degradation that can occur during transport, particularly in complex matrices like wastewater where inhibitors and degradative enzymes are abundant. In addition, the approach enables testing in locations without reliable access to centralised laboratory facilities, expanding WBE coverage to underserved areas.

The evaluation of our method with raw wastewater samples from active surveillance programs is an important step towards practical effectiveness in real-world near-source settings. We believe the

significant advantages in speed, cost, and accessibility justify the near-source approach, even without concerns about amplified DNA stability. Based on the Reviewer's comments, we have revised the manuscript to more clearly articulate the rationale for near-source testing. We have added explanatory text in the Introduction and Discussion sections to highlight the practical advantages beyond nucleic acid stability.

Amendments to our manuscript:

Main Text Lines 70-78: " While conventional WBE requires sample transportation and centralised laboratory testing, near-source approaches offer significant advantages beyond just preserving nucleic acid integrity (Fig. 1a). These include dramatically reduced turnaround times (>24-72 hours) for public health decision-making, lower overall testing costs by eliminating transportation and complex laboratory infrastructure, and expanded accessibility in locations with limited laboratory resources, such as wastewater pumping stations, schools, hospitals, airports, prisons, or care homes in developed and developing countries³²⁻³⁴."

Main Text Lines 360-366: "As a result, the lab-in-a-suitcase format enables complete on-site processing - from sample collection to result interpretation - without requiring sample transportation or specialised laboratory facilities. This approach addresses several key limitations of conventional WBE in turnaround time and high costs and extends WBE capabilities to underserved or remote locations. While amplified DNA stability is not a significant concern, the operational advantages of integrated near-source testing justify this approach for rapid, accessible wastewater surveillance."

Main Text Lines 431-438: "In future, we aim to field-test the 'lab-in-a-suitcase' concept with diverse end-users, developing standardised operating procedures (SOP), that include quality control protocols such as preserving test strips for laboratory verification or performing replicate testing to ensure reproducibility. The SOPs would incorporate standardised freeze-dried reagent kits with built-in controls to minimise variability across testing sites, while also exploring the integration of readers, modulation and smartphones with data connectivity for real-time geospatial data collection, automated analysis and potential remote quality control oversight."

Reviewer #2 Comment 7: The manuscript states that the RPA-dipstick-CBNP assay requires 40 minutes of amplification, whereas the RPA-dipstick-FND assay requires only 25 minutes. However, it is not explicitly discussed why such a large difference in amplification time exists between the two approaches. The authors should clarify whether this difference is due to fundamental differences in signal generation, nanoparticle interaction, or fluorescence readout sensitivity. If the amplification time was optimized differently for each assay, a brief justification should be provided.

In addition, this study employs two different RNA extraction methods: PEG precipitation and magnetic bead extraction. However, the rationale for selecting each method is not clearly explained. Additionally, PEG precipitation is used for the CBNP assay, while magnetic bead extraction is applied to the FND assay, but there is insufficient discussion on how this difference affects the limit of detection (LoD) and sensitivity. I strongly recommend that the authors clearly justify the choice of extraction methods for each assay and discuss whether these differences influence the study's conclusions. Furthermore, experimental data or references should be provided to demonstrate that the choice of RNA extraction method does not bias the results.

Response:

We thank the Reviewer for highlighting the need for clarification regarding amplification time differences and extraction method choices. We have addressed both aspects below:

1. Amplification Time Differences: We apologise for not clearly explaining the rationale behind the different amplification times for CBNP (40 min) versus FND (25 min) assays. As shown in Fig. S5a, our optimisation studies revealed a substantial sensitivity difference between these detection methods. The FND readout demonstrated a 15,375-fold improvement in sensitivity compared to CBNP when tested with identical ssDNA model amplicon on LFTs. This dramatic sensitivity enhancement allows the FND method to detect much lower amplicon concentrations. Our evaluation of RPA kinetics (Fig. S5b) showed approximately 10,000-fold amplification within 20 minutes for high concentration target strands. Based on these findings, we determined that 25 minutes of amplification provides sufficient DNA yield for reliable FND detection, even with lower starting concentrations, while the less sensitive CBNP method requires 40 minutes to achieve adequate signal. The 25-minute protocol balances sensitivity and rapid turnaround time, which is particularly important for near-source applications.

Fig.S5 Development and optimisation of nanodiamond-enhanced lateral flow assay. a) LoD comparison between FND and CBNP based on ssDNA model amplicon on LFT. The assay readout sensitivity improvement of 15375-fold over CBNPs to FNDs. b) Evaluation of converted ssDNA copy numbers over the RPA amplification based on the high concentration of target strands. This indicates around a 10,000-fold change in DNA copies over RPA in 20 minutes. 25 minutes of RPA amplification can be applied for FND assay.

2. RNA Extraction Methods: We apologise for any confusion regarding extraction methods. To clarify, both assay types (CBNP and FND) were evaluated using the same magnetic bead extraction protocol for the 62 wastewater samples, ensuring a direct comparison of detection performance across different sensing nanoparticles. As shown in Fig. S6c (please also see the response to the Reviewer #1 Comment 3), we validated our magnetic bead method against conventional PEG precipitation (the benchmark method used in national WBE surveillance programs) and found comparable RNA recovery rates between the two approaches. The magnetic bead method was selected for our near-source application because it offers similar recovery efficiency while significantly reducing processing time and complexity, making it suitable for field deployment. In addition, a PEG precipitation sample (with RNA concentration checked) was used as a standard wastewater for the developing of the RPA assay, to mimic the wastewater sensing sample status in the early developing stage, which were mentioned in main text lines 202-204.

We have revised the manuscript to clarify these methodological choices and their rationales, enhancing the Methods section with additional details on the optimisation process for both amplification times and extraction methods.

Amendments to our manuscript:

Main Text Lines 246-251: "We found that only 25 mins RPA was required (compared to 40 mins with CBNP) marking a 37.5% reduction in assay time. The assay readout sensitivity improvement of 15375-fold over CBNPs to FNDs means that we can amplify for just 25 minutes to achieve higher enough sensitivity using FND dipsticks (Fig. S5, and see Methods)."

Methods Lines 592-599: "The amplification time was optimised differently for CBNP and FND detection methods based on their sensitivity differences. As demonstrated in Fig. S4, FND detection is approximately 15,375-fold more sensitive than CBNP when using identical amplicon concentrations. This substantial sensitivity enhancement allows the FND method to reliably detect amplicons after just 25 minutes of RPA, while the CBNP method requires 40 minutes to achieve adequate signal strength. This optimization balances detection sensitivity with the need for rapid turnaround in near-source applications."

Main Text Lines 190-193: "The FND readout was demonstrated a 15,375-fold improvement in sensitivity compared to CBNP when tested with identical ssDNA model amplicon on LFTs."

Main Text Lines 190-193: "62 wastewater samples were evaluated by this assay. To ensure direct comparability between CBNP and FND detection methods, all 62 wastewater samples were processed using this identical magnetic bead extraction protocol (will be detailed discussed later)²⁶."

Methods Lines 602-607: "To assess extraction efficiency, we conducted a comparative analysis between our magnetic bead method and conventional PEG precipitation across 30 raw wastewater samples. Both methods were evaluated using identical qPCR primers and conditions. The magnetic bead extraction demonstrated comparable recovery efficiency (two-tailed t -test $P=0.2254$, $t=1.239$), shown in Fig.S7c. This comparison confirms that our streamlined magnetic bead extraction provides suitable recovery rates for reliable near-source detection while significantly reducing processing complexity and time."

Supplementary Information Fig.S7c Line 84-97: (please refer to the Reviewer #1 Comment 3)

Reviewer #3:

Reviewer #3 Overall comments: The authors have developed and tested a number of rapid dipstick style assays for the detection of SARS-CoV-2 in wastewater, comparing the sensitivity and specificity of detection with RT-qPCR. This is a strong scientific advancement, given the current limitations around transporting wastewater samples to laboratories before analysis and reporting. The experiments and analyses are well designed and well reported. I have a few suggestions and questions, primarily around the framing

Response:

We thank the Reviewer for highlighting our 'strong scientific advancement' and that 'our experiments and analyses are well designed and well reported'.

Reviewer #3 Comment 1: Abstract, Line 20 – "RT-qPCR" might be too prescriptive. I would just say PCR as many programs have moved to digital droplet PCR.

Response:

We thank the Reviewer for this suggestion. We have replaced "RT-qPCR" with "PCR", acknowledging the diversity of PCR-based approaches currently used in WBE programs.

Amendments to our manuscript:

Main Text Lines 20-21: "However, gold-standard PCR necessitates transporting samples to laboratories, with significant reporting delays....."

Reviewer #3 Comment 2: Line 50 – switch monkeypox to mpox.

Response:

We thank the suggestion from Reviewer. We have updated the term from "monkeypox" to "mpox" to reflect current terminology.

Amendments to our manuscript:

Main Text Lines 43-45: "Today, more than 70 countries have adopted Wastewater-Based Epidemiology (WBE) an early warning tool for COVID-19, mpox, and many other pathogens¹⁰⁻¹³."

Reviewer #3 Comment 3: Lines 73-74 – digital droplet PCR is the gold standard in the US. Cultured virus is the gold standard for polio environmental surveillance.

Response:

We thank the suggestion from Reviewer. We have revised our sentences of gold standard methods to acknowledge reviewer's advice. Relevant references were added to represent the ddPCR as the emerging gold standard in US-based programs and cultured virus as the gold standard for polio environmental surveillance.

Amendments to our manuscript:

Main Text Lines 67-70: "The current 'gold-standard' WBE method of PCR (digital droplet PCR, ddPCR, in the US, or reverse-transcription quantitative PCR, RT-qPCR) requires sample collection, cold-chain

transportation to laboratory, and specialised equipment with staff for sample processing, concentration or virus cultivation (for polio), and performing PCR²⁶⁻³⁰.”

Reviewer #3 Comment 4: The authors compare sensitivity and specificity of their dipsticks to RT-qPCR and on the model pathogen SARS-CoV-2. Digital droplet PCR is much more sensitive to detect pathogens in wastewater, and at least the US-based programs are all switching to DD-PCR. It would be important to mention this limitation in the discussion. Additionally, SARS-CoV-2 is very abundant in wastewater – much more so than polio, influenza, and RSV. The only other pathogen that gets similar detection rates is norovirus. This will have implications for generalizing the technology to other pathogens. For example, in NY we estimated that polio surveillance in wastewater was much less sensitive than SARS-CoV-2 surveillance. For example, see: Larsen DA, Hill D, Zhu Y, Alazawi M, Chatila D, Dunham C, Faruolo C, Ferro B, Godinez A, Hanson B, Insaf T. Non-detection of emerging and re-emerging pathogens in wastewater surveillance to confirm absence of transmission risk: A case study of polio in New York. PLOS global public health. 2024 Dec 31;4(12):e0002381.

Response:

We thank the Reviewer for this valuable comment and for suggesting the relevant reference. We have revised our manuscript to acknowledge that digital droplet PCR (ddPCR) is increasingly becoming the standard for wastewater surveillance in the US, particularly for its superior quantitative capabilities in applications such as antimicrobial resistance monitoring. We agree that SARS-CoV-2 generally exhibits higher abundance in wastewater compared to pathogens like polio, influenza, and RSV, mentioned by the Reviewer, which presents additional challenges for detection. While our study focused on SARS-CoV-2 as a proof-of-concept, we acknowledge the limitations when extending this technology to less abundant pathogens. Based on our expertise on WBE, we believe the detection sensitivity in WBE applications relies on effective sample concentration and RNA extraction protocols that maximise recovery from original wastewater samples. The RT-qPCR assay referenced in our study (Farkas et al., 2022, national WBE surveillance program)¹¹ demonstrated high sensitivity with a limit of detection of 0.92 genome copies/ μ L RNA extract, corresponding to 611 genome copies/L, gc/L in wastewater, which is comparable to many ddPCR assays recently published. Our FND method's theoretical sensitivity of 140 gc/L in original wastewater (assuming optimal RNA recovery) indicates the potential of quantum sensing for various pathogens across a range of abundance levels. For less abundant pathogens, our method would require further optimisation to achieve reliable detection. We recognise this limitation and have outlined approaches to address it in future work. Specifically, we would focus on enhancing RNA recovery efficiency from complex wastewater matrices, which is particularly critical for low-abundance targets. Additionally, as demonstrated in our previous study (Farkas et al., 2022)¹¹, increasing sample volumes from 50 mL to 100-150 mL significantly improves detection of rare targets. This simple modification, combined with our sensitive FND detection platform, could extend the utility of our approach to a broader range of pathogens with varying abundance levels in wastewater.

We have revised the manuscript to address the Reviewer's points by adding discussion of these limitations, acknowledging the transition to ddPCR as a standard protocol for WBE in the US, and outlining our future development and optimisation plans. We have also cited the suggested Larsen *et al.* (2024) reference, which provides valuable context regarding the challenges of detecting less abundant pathogens like polio in wastewater surveillance programs.

Amendments to our manuscript:

Main Text Lines 438-447: “In addition, we also noticed that SARS-CoV-2 is typically present at higher concentrations in wastewater compared to pathogens such as polio⁵⁴. Future developments would focus

on enhancing RNA recovery efficiency from complex wastewater matrices - a critical factor for low-abundance targets. As demonstrated in our previous work⁴⁹, a straightforward approach involves increasing sample volumes from 50 mL to 100-150 mL, which significantly improves detection of rare targets. This modification, combined with the high sensitivity of our FND detection platform, could extend the utility of our approach to a broader range of pathogens with varying abundance levels in wastewater, while maintaining the advantages of rapid, field-deployable testing.”

References

1. Cherkaoui, D., Huang, D., Miller, B. S., Turbé, V. & McKendry, R. A. Harnessing recombinase polymerase amplification for rapid multi-gene detection of SARS-CoV-2 in resource-limited settings. *Biosens Bioelectron* **189**, 113328 (2021).
2. Miller, B. S. *et al.* Spin-enhanced nanodiamond biosensing for ultrasensitive diagnostics. *Nature* **587**, 588–593 (2020).
3. Miller, B. S. *et al.* Quantifying Biomolecular Binding Constants using Video Paper Analytical Devices. *Chemistry – A European Journal* **24**, 9783–9787 (2018).
4. Wade, M. J. *et al.* Understanding and managing uncertainty and variability for wastewater monitoring beyond the pandemic: Lessons learned from the United Kingdom national COVID-19 surveillance programmes. *J Hazard Mater* **424**, 127456 (2022).
5. Farkas, K. *et al.* Concentration and Quantification of SARS-CoV-2 RNA in Wastewater Using Polyethylene Glycol-Based Concentration and qRT-PCR. *Methods Protoc* **4**, 1–9 (2021).
6. Hamilton, K. A., Wade, M. J., Barnes, K. G., Street, R. A. & Paterson, S. Wastewater-based epidemiology as a public health resource in low- and middle-income settings. *Environmental Pollution* **351**, 124045 (2024).
7. Ofori, B. *et al.* Leveraging wastewater-based epidemiology to monitor the spread of neglected tropical diseases in African communities. *Infect Dis (Lond)* **56**, 697–711 (2024).
8. Miller, B. S. *et al.* Sub-picomolar lateral flow antigen detection with two-wavelength imaging of composite nanoparticles. *Biosens Bioelectron* **207**, 114133 (2022).
9. Holstein, C. A., Griffin, M., Hong, J. & Sampson, P. D. Statistical Method for Determining and Comparing Limits of Detection of Bioassays. *Anal Chem* **87**, 9795–9801 (2015).
10. Karthikeyan, S. *et al.* High-Throughput Wastewater SARS-CoV-2 Detection Enables Forecasting of Community Infection Dynamics in San Diego County. *mSystems* **6**, (2021).
11. Farkas, K. *et al.* Comparative Assessment of Filtration- and Precipitation-Based Methods for the Concentration of SARS-CoV-2 and Other Viruses from Wastewater. *Microbiol Spectr* **10**, (2022).